# Quantifying portable genetic effects and improving cross-ancestry genetic prediction with GWAS summary statistics

Jiacheng Miao[1,6], Hanmin Guo [2,6], Gefei Song[1], Zijie Zhao[1], Lin Hou [2,3,7] & Qiongshi Lu [1,4,5,7]

Polygenic risk scores (PRS) calculated from genome-wide association studies (GWAS) of Europeans are known to have substantially reduced predictive accuracy in non-European populations, limiting their clinical utility and raising concerns about health disparities across ancestral populations. Here, we introduce a statistical framework named X-Wing to improve predictive performance in ancestrally diverse populations. X-Wing quantifies local genetic correlations for complex traits between populations, employs an annotation-dependent estimation procedure to amplify correlated genetic effects between populations, and combines multiple population-specific PRS into a unified score with GWAS summary statistics alone as input. Through extensive benchmarking, we demonstrate that X-Wing pinpoints portable genetic effects and substantially improves PRS performance in non-European populations, showing 14.1%–119.1% relative gain in predictive $R^2$ compared to state-of-the-art methods based on GWAS summary statistics. Overall, X-Wing addresses critical limitations in existing approaches and may have broad applications in cross-population polygenic risk prediction.

Genome-wide association studies (GWAS) have identified tens of thousands of genotype-phenotype associations for human complex traits[1,2]. Polygenic risk score (PRS) based on GWAS, typically calculated as a weighted sum of trait-associated allele counts across numerous loci in the genome, is an effective tool to quantify the aggregated genetic propensity for a trait or disease[3–8]. With rapid advances in GWAS sample size and statistical methodology for modeling summary-level data, PRS has shown substantially improved prediction accuracy and great potential in disease risk screening and precision medicine[9–11]. However, since the vast majority of GWAS participants are of European descent, current PRS models are more effective in Europeans but are known to have substantially reduced accuracy in other populations, which severely limits their clinical utility[12–16]. There is an urgent need to improve the effectiveness of PRS in diverse human populations and provide equitable access to genomic advances in precision medicine[14,17–20].

There have been three types of approaches to improve cross-ancestry genetic prediction in the literature. First, prioritizing causal variants using functional genomic annotations can improve the portability of PRS based on European GWAS[21–23]. Second, several studies combine multiple PRS trained in various populations using linear regression to optimize the predictive performance in the target (non-European) population[16,23,24]. The third type of approach parametrizes the degree to which genetic effects are correlated across populations, and integrates GWAS summary statistics from multiple populations in a multivariate model to improve effect size estimation and prediction

[1]Department of Biostatistics and Medical Informatics, University of Wisconsin–Madison, Madison, WI 53706, USA. [2]Center for Statistical Science, Department of Industrial Engineering, Tsinghua University, Beijing 100084, China. [3]MOE Key Laboratory of Bioinformatics, School of Life Sciences, Tsinghua University, Beijing 100084, China. [4]Department of Statistics, University of Wisconsin–Madison, Madison, WI 53706, USA. [5]Center for Demography of Health and Aging, University of Wisconsin–Madison, Madison, WI 53706, USA. [6]These authors contributed equally: Jiacheng Miao, Hanmin Guo. [7]These authors jointly supervised this work: Lin Hou, Qiongshi Lu. ✉e-mail: houl@tsinghua.edu.cn; qlu@biostat.wisc.edu

accuracy in each respective population[16,25–27]. These models have achieved moderately improved predictive performance compared to conventional single-population approaches, but several critical limitations and challenges remain. First, previous studies used epigenetic regulatory annotations to prioritize variants for PRS[21–23]. While these annotations improved PRS portability for some traits, they are not designed to quantify the correlated genetic effects between populations[28], and there is no guarantee that the same set of annotations will improve PRS performance for all complex traits. Additionally, existing statistical frameworks that leverage functional annotation data to improve PRS[29–33] do not apply to multi-ancestry predictive modeling. Finally, in order to combine multiple population-specific PRS, the current practice requires additional data from the target (non-European) population. This includes individual-level genotype and phenotype samples that are independent of the GWAS used to train single-population PRS. In practice, this type of data can be nearly impossible to obtain[34]. In order to have broad applications, PRS models need to use the increasingly accessible GWAS summary statistics from global populations[35–37] as input.

In this work, we introduce a cross-population weighting (X-Wing) framework for genetic prediction. There are three main innovations in our approach. First, we introduce an annotation framework based on cross-population local genetic correlation. This annotation extends our previous work[38] to directly quantify correlated (portable) genetic effects between multiple ancestral populations. Second, we introduce a Bayesian method to incorporate functional annotation data into multi-population PRS modeling, where annotation-dependent statistical shrinkage amplifies the effects of annotated variants (*i.e.*, variants with correlated effects between populations). Finally, we resolve a long-standing challenge in the field and introduce a method to combine multiple PRS trained in various populations using GWAS summary data alone as input. We demonstrate the superior performance of X-Wing PRS through extensive benchmarking using numerous GWAS datasets, including UK Biobank (UKB)[39], Biobank Japan (BBJ)[40], and Population Architecture using Genomics and Epidemiology Consortium (PAGE) study[41].

## Results
### Methods overview
The X-Wing workflow is illustrated in Fig. 1. We have previously developed a scan statistic approach[38] for identifying genomic regions with correlated effects on two complex traits. In this paper, we first extend this approach to identify correlated genetic effects on the same trait between two populations. Once identified, these

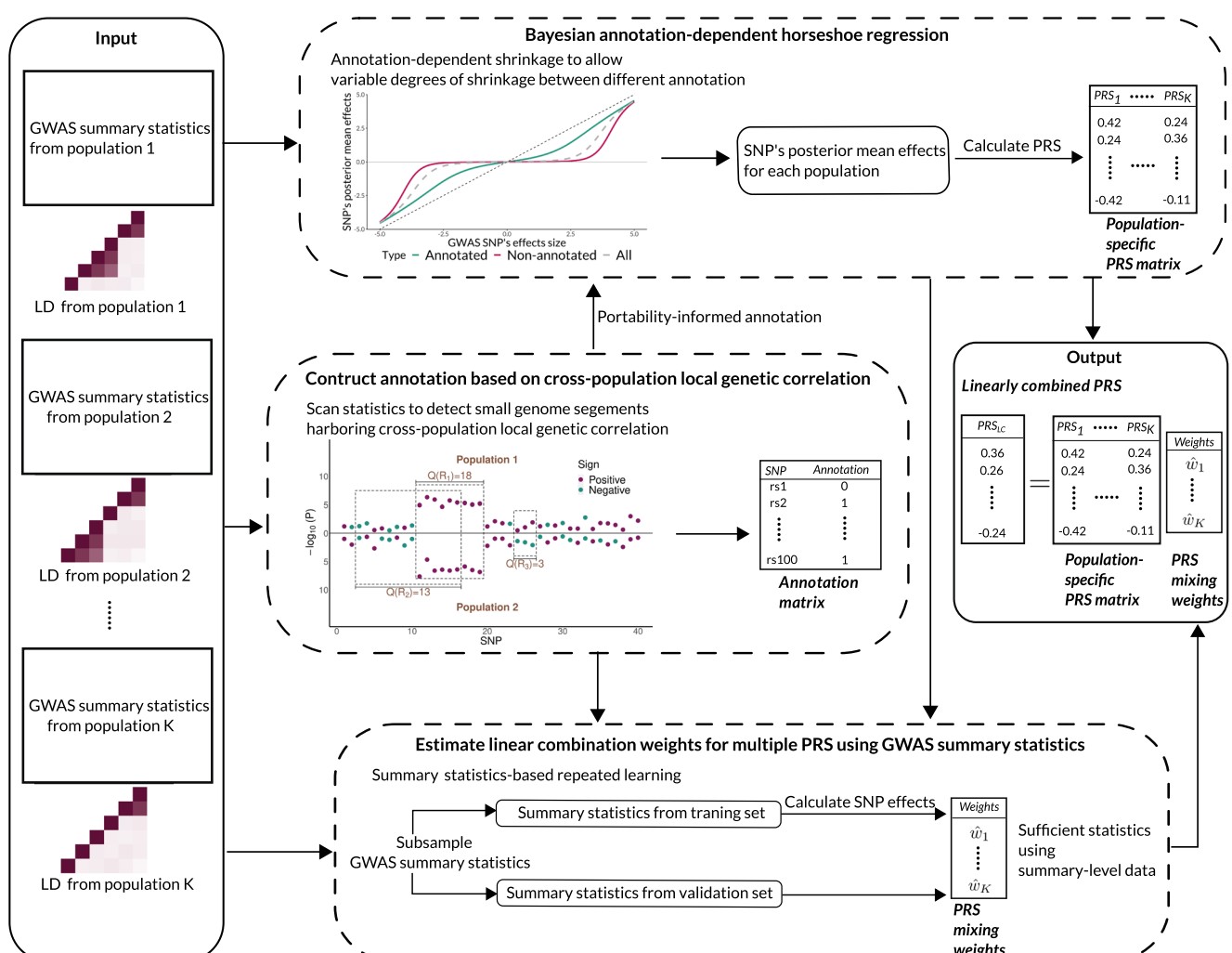

**Fig. 1 | X-Wing workflow.** X-Wing uses GWAS summary statistics and population-matched LD references as input. It first employs a scan statistic approach to detect genome segments showing local genetic correlation between populations. Next, it incorporates the local genetic correlation annotation into a Bayesian PRS model, amplifying SNP effects that are correlated between populations. Finally, it uses summary statistics-based repeated learning to combine multiple population-specific PRS and produce the final PRS with improved accuracy.

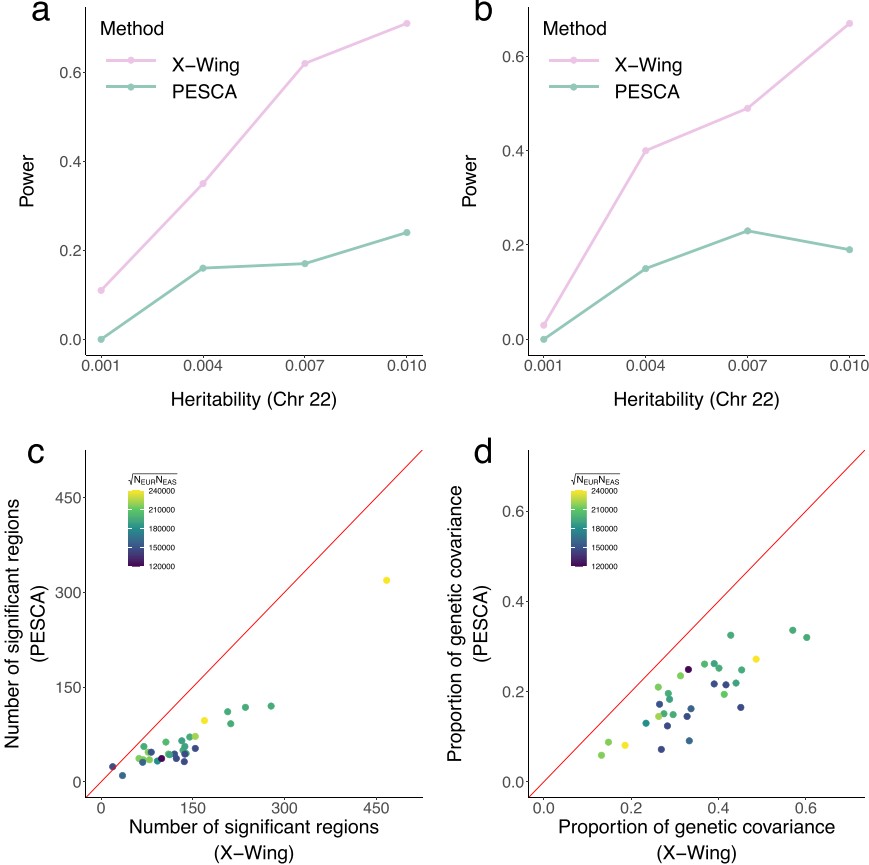

**Fig. 2 | X-Wing achieves superior statistical power in identifying cross-population local genetic correlation. a**, **b** Statistical power in simulations under a heritability enrichment framework. Power is defined as the proportion of simulation repeats that the true signal region is identified. Panels (**a**) and (**b**) illustrate results for continuous and binary trait outcomes, respectively. **c** Number of regions with significant cross-population genetic correlations identified by X-Wing and PESCA for 31 complex traits. **d** Proportion of total genetic covariance explained by significant local regions for 31 complex traits. Genetic covariance measures covariance of additive genetic component between two populations. In both panels (**c**) and (**d**), GWAS sample sizes are indicated by the color of each data point, and the diagonal line is highlighted in red.

genomic regions explain the shared genetic basis of the phenotype between populations and could be an informative annotation for prioritizing single-nucleotide polymorphisms (SNPs) in PRS models. Next, to quantitatively incorporate this annotation in multipopulation PRS modeling, we introduce a Bayesian framework in which annotation-dependent shrinkage parameters allow variable degrees of statistical shrinkage between annotated and non-annotated SNPs. Coupled with other shrinkage parameters that do not depend on functional annotations, this framework amplifies SNP predictors that show correlated effects between populations while ensuring robustness to diverse types of genetic architecture[42–45]. Although we only explore its performance using the annotation derived from local genetic correlation in this paper, we note that this is a general framework that allows an arbitrary collection of annotation variables as input and also accounts for population-specific linkage disequilibrium (LD) and allele frequencies. Finally, we introduce an innovative strategy to linearly combine multiple PRS trained in different populations using summary association data alone. We employ a summary statistics-based repeated learning approach motivated from our recent work[8] and its extension[33] to estimate the regression weights for combining multiple PRS. The entire X-Wing procedure only requires GWAS summary data and LD references as input, which is a major advance compared to existing approaches. We present the statistical details and technical discussions in Methods and Supplementary Methods.

## X-Wing pinpoints local genetic correlation between ancestral populations

We first carried out simulations to assess the performance of our approach in identifying cross-population local genetic correlations. Using European and East Asian samples in 1000 Genomes Project phase III data[46], we simulated chromosome 22 genotypes of 50,000 individuals, and simulated quantitative traits in two populations under an infinitesimal model with varying heritability levels (Methods). When the traits in two populations are independent, X-Wing showed well-controlled type-I error rates (Supplementary Data 1). Since no existing method can estimate local genetic correlation between two distinct ancestral populations, we compared our results with PESCA[47], a recently developed approach for estimating the risk SNP proportion shared by two populations, to gain some perspective on the statistical property of our inference results. PESCA also showed well-controlled type-I error across simulation settings, but X-Wing consistently achieved higher statistical power, especially when heritability is large (Fig. 2a).

To assess the robustness of our method to model mis-specification, we considered additional data-generating models in which SNP heritability is enriched in certain genomic regions[38] or is dependent on LD and minor allele frequency (MAF)[48]. We also investigated binary phenotypes using a liability threshold model. We obtained consistent results in these analyses, with our method showing well-controlled type-I error (Supplementary Data 2–4) and superior statistical power (Fig. 2b and Supplementary Fig. 1).

As a robustness check, we also performed simulations based on genome-wide data. X-Wing showed well-calibrated type-I error rates (Supplementary Data 5) and identified more signal regions than PESCA when two populations shared local genetic correlations (Supplementary Fig. 2). Notably, PESCA suffered substantial type-I error inflation when two simulated traits are independent (Supplementary Data 5) and showed high false positive rates when two populations are correlated (Supplementary Data 6).

## Local genetic correlation between Europeans and East Asians for 31 traits

We estimated local genetic correlations for 31 complex traits (Supplementary Data 7) between Europeans and East Asians using GWAS summary statistics from UKB ($N$ = 314,921-360,388)[39] and BBJ ($N$ = 42,790-159,095)[40]. In total, we identified 4160 regions with significant cross-population local genetic correlations across 31 traits (FDR < 0.05; Supplementary Data 8). Of these, the vast majority (4,008 regions) showed positive correlations. 958 identified regions have genome-wide significant SNPs in both populations and 2,119 have significant SNPs in only one population (Supplementary Fig. 3). The number of significantly correlated regions identified for each trait pair is proportional to the global genetic correlations estimated from genome-wide data[25] (Supplementary Fig. 4; correlation $r$ = 0.49). As a comparison, we also applied PESCA to these data, and identified 1,968 risk regions shared by two populations (Supplementary Data 8). Our approach identified more significant regions in 30 out of 31 traits (Fig. 2c). The regions identified by our approach also explained larger proportions of cumulative genetic covariance in all 31 traits (Fig. 2d). Further, all conclusions remained similar when only HapMap3 SNPs were included in the analysis (Supplementary Fig. 5).

Overall, regions with significant local genetic correlations cover 0.06% (basophil) to 1.73% (height) of the genome, but explain 13.22% (diastolic blood pressure) to 60.17% (mean corpuscular volume) of the total genetic covariance between Europeans and East Asians (Fig. 3a and Supplementary Data 9), showing fold enrichments ranging from 28.09 to 546.83. Cross-population genetic correlations inside X-Wing-identified regions are substantially higher than the genome-wide genetic correlation estimates, while correlations in the remaining genome are consistently lower (Fig. 3b). Notably, among the traits we analyzed, basophil count has the lowest cross-population genetic correlation ($r_g$ = 0.23) which is consistent with previous reports[49,50]. But even for basophil count, we observed a substantial genetic correlation in regions identified by our approach ($r_g$ = 0.83). To guard against statistical artifacts, we performed falsification tests by simulating a trait that is uncorrelated between populations (Methods). We did not identify significant global or local correlations for this simulated trait (Fig. 3b).

We also sought to replicate local correlations between Europeans and East Asians for four lipid traits (HDL cholesterol, LDL cholesterol, total cholesterol, and triglycerides) in independent data. We used European GWAS from the Global Lipids Genetics Consortium (GLGC, $N$ = 95,454-100,184)[51] and East Asian GWAS from the Asian Genetic Epidemiology Network (AGEN, $N$ = 27,657-34,374)[52] as the replication datasets (Supplementary Data 10). In total, we identified 124 significant regions for four lipid traits in the replication analysis. 102 of them overlapped with significant regions identified in the discovery stage (Fig. 3c). Regions identified in the discovery stage showed substantial enrichment for genetic covariance in the replication data (greater than 100-fold for all four traits; Supplementary Data 11). Further, we ranked the regions identified in the discovery stage by their p-values. The cumulative proportion of genetic covariance explained by these regions were nearly identical between discovery and replication analyses (Fig. 3d and Supplementary Fig. 6).

## Local genetic correlation annotation improves PRS prediction accuracy across populations

Next, we investigated whether incorporating the annotation based on local genetic correlation can improve the cross-ancestry prediction accuracy of PRS. We used European GWAS from UKB and East Asian GWAS from BBJ to train PRS for 31 complex traits, and evaluated PRS performance using independent East Asian samples in UKB ($N$ = 2683). In this analysis, our approach jointly models GWAS in two populations and outputs separate SNP weights for Europeans and East Asians (Methods). Here, we used annotation-informed PRS based on posterior SNP effects estimated for Europeans, and report its performance in the East Asian target sample (thus, quantifying the portability of European scores in the East Asian population). PRS performance is quantified using partial $R^2$ adjusting for covariates (Methods). Our annotation-informed PRS showed a 4.6% ($P_{wilcoxon}$ = 7.0e-6) and 35.2% ($P_{wilcoxon}$ = 1.0e-7) median relative improvement in $R^2$ compared to PRS-CSx[14] and XPASS[20] (Fig. 4a; Supplementary Fig. 7; Supplementary Data 12), demonstrating the effectiveness of incorporating local genetic correlation annotation. In fact, we found both higher overall $R^2$ and larger increase of $R^2$ in annotated genomic regions (i.e., regions with correlated effects between populations) using our approach. PRS using only SNPs outside annotated regions did not show any improvement (Fig. 4b, c and Supplementary Data 13). We also compared our results with PolyFun-pred[18], an approach that uses functional fine-mapping to improve PRS performance. Our PRS showed a substantial 78.1% ($P_{wilcoxon}$ = 5.8e-4) relative gain in $R^2$, suggesting that fine-mapping in European population alone is a sub-optimal approach compared to multi-population joint modeling (Supplementary Fig. 8 and Supplementary Data 12).

## X-Wing combines multiple population-specific PRS using GWAS summary statistics

Next, we investigated the benefit of combining multiple PRS trained for different populations into a single score. We evenly split the East Asian target sample in UKB into a validation set in which we fit a regression model to combine the European and East Asian scores, and a testing set in which we evaluate the performance of combined PRS. We compared the prediction accuracy of X-Wing PRS with PRS-CSx, XPASS, and PolyPred+ using the same regression approach to combine scores. X-Wing showed an median $R^2$ relative increase of 3.9% ($P_{wilcoxon}$ = 1.0e-6), 46.1% ($P_{wilcoxon}$ = 1.9e-9), and 24.7% ($P_{wilcoxon}$ = 0.02) compared to PRS-CSx, XPASS, and PolyPred+ in East Asian target samples, respectively (Fig. 5a, Supplementary Fig. 7, and Supplementary Data 12). We also assessed the combined scores based on UKB, BBJ, and PAGE in admixed Americans and Africans. Our method showed a 3.2% ($P_{wilcoxon}$ = 0.01) and 1.9% ($P_{wilcoxon}$ = 0.01) median relative increase in $R^2$ compared to PRS-CSx in admixed Americans and Africans, respectively (Supplementary Figs. 9, 10 and Supplementary Data 14, 15). XPASS was excluded since it cannot take more than two GWAS datasets as input and PolyPred+ was also excluded since it did not release PRS coefficients estimated using PAGE. We also performed sensitivity analyses by varying the size of genetic correlation annotation, upper bound of region size, and merge distance in identifying local genetic correlations. We also examined PRS performance after excluding the MHC region and explored estimating the global shrinkage parameter using a model tuning approach instead of the full Bayesian procedure (Supplementary Methods). We obtained consistent results in these analyses, demonstrating the robustness of X-Wing to these choices (Supplementary Figs. 11–18, Supplementary Data 16–22). We also performed simulations to benchmark the predictive performance of PRS using X-Wing, PRS-CSx and XPASS (Supplementary Methods). X-Wing shows consistent improvement over PRS-CSx and XPASS in the presence of local genetic correlation across two populations (Supplementary Fig. 19).

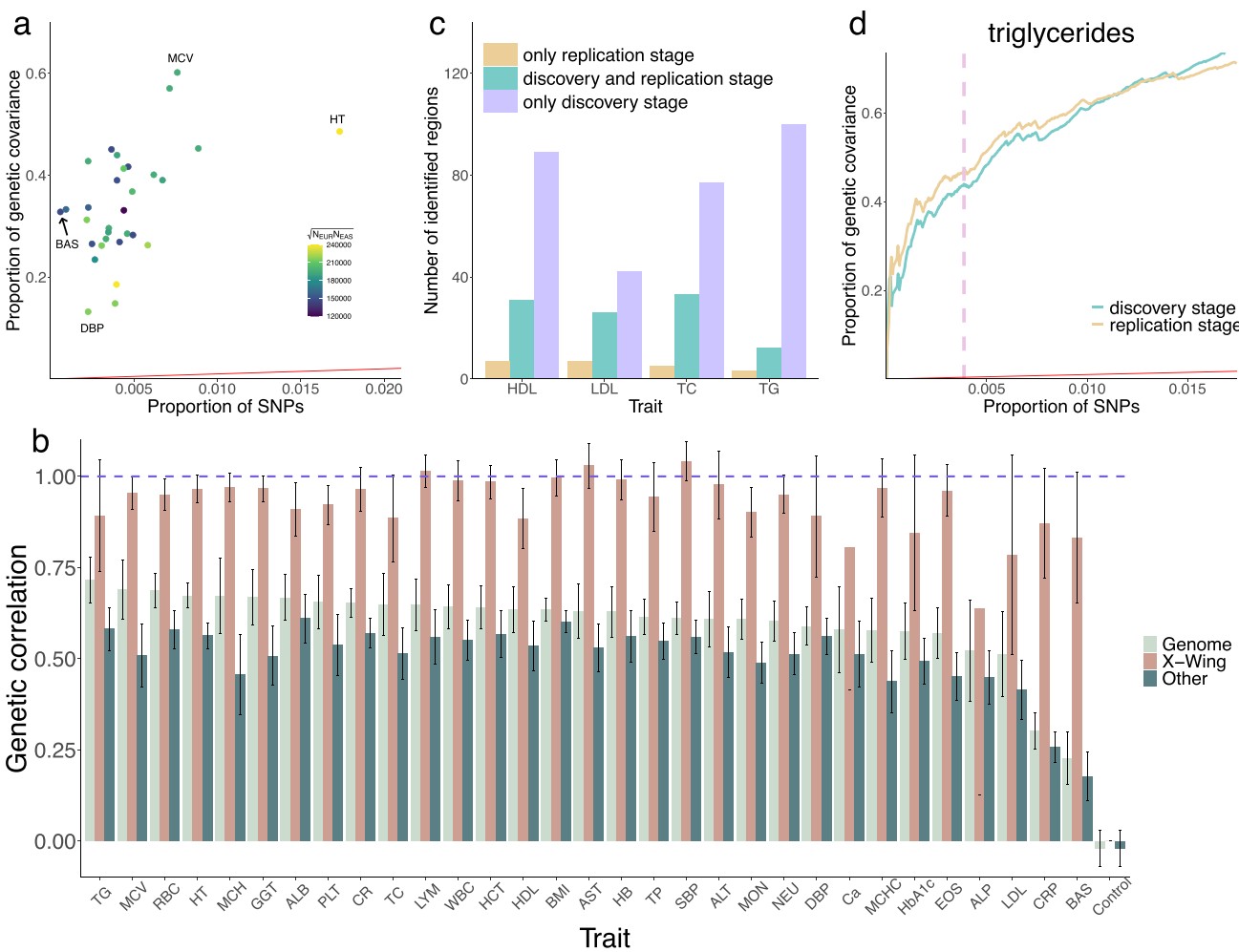

**Fig. 3 | X-Wing identifies genomic regions strongly enriched for correlated genetic effects between Europeans and East Asians. a** Scatter plot shows the proportion of SNPs in regions identified by X-Wing and the proportion of cross-population genetic covariance explained by these SNPs. All data points are above the diagonal line highlighted in red, showing substantial enrichment. **b** Cross-population genetic correlation for 31 complex traits. Three bars denote the global genetic correlation estimated from genome-wide data (light green), genetic correlation in regions identified by X-Wing (brown), and genetic correlation outside regions identified by X-Wing (dark green). Results for a simulated uncorrelated trait are labeled as 'Control'. All traits are ordered according to the global genetic correlation estimates. Error bars indicate 95% confidence interval. The centre for the error bars represents the point estimates for genetic correlation. A list of trait acronyms can be found in Supplementary Data 7. **c** Bar plot shows the number of significant regions identified only in discovery stage (purple), only in replication stage (orange), and in both stages (blue) for four lipid traits. HDL, LDL, TC, TG stand for HDL cholesterol, LDL cholesterol, total cholesterol, and triglycerides, respectively. **d** Cumulative proportion of genetic covariance explained by regions identified in the discovery stage for triglycerides. Analogous results for HDL cholesterol, LDL cholesterol, and total cholesterol are shown in Supplementary Fig. 6. Pink dashed line indicates FDR cutoff of 0.05. Red line represents the diagonal line of y = x. Genetic correlation and genetic covariance were calculated using XPASS.

Finally, we demonstrated that population-specific PRS can be combined using GWAS summary data alone. We used summary-statistics-based repeated learning (Methods), instead of regressions trained on reserved samples, to linearly combine multiple PRS. This analytic strategy showed almost identical results compared to the gold-standard regression approach in both East Asian, admixed American, and African target samples (regression slope = 0.983, 1.007, and 0.971) (Fig. 5b, Supplementary Figs. 10, 20, and Supplementary Data 23). Notably, if no external individual-level data are available for regression model training, the current best PRS approach in practice is to use posterior SNP effects estimated for one population (Methods). Compared to the best-performing population-specific scores, X-Wing PRS can be trained using the same input data but showed a substantial improvement in prediction accuracy, with the median relative increase of $R^2$ ranging from 25.4 to 58.5% ($P_{wilcoxon} = 1.3e-8$ to $1.9e-9$) in East Asians, 14.1–74.2% ($P_{wilcoxon} = 4.8e-4$ to $2.4e-4$) in admixed Americans, and 30.2–119.1% ($P_{wilcoxon} = 0.01–2.4e-4$) in Africans (Fig. 5c and

Supplementary Figs. 10, 20, 21). We further compared X-Wing performance with the "-meta" option in PRS-CSx that requires no additional validation cohort. X-Wing showed a median $R^2$ relative increase of 10.2% ($P_{wilcoxon} = 3.6e-3$), 9.6% ($P_{wilcoxon} = 0.02$), and 20.2% ($P_{wilcoxon} = 2.4e-4$) for traits in East Asians, Africans, and admixed Americans, respectively (Supplementary Fig. 22). We also evaluated X-Wing performance using a binary trait, type-2 diabetes, in East Asians. X-Wing PRS showed both higher liability $R^2$ and AUC over PRS-CSx and XPASS (Supplementary Fig. 23)[53,54]. Overall, X-Wing PRS shows better predictive performance over alternative methods tested (Supplementary Fig. 24).

## Discussion

In this paper, we introduced X-Wing, a sophisticated statistical framework for improving PRS performance in ancestrally diverse populations. X-Wing quantifies cross-population local genetic correlation, and incorporates it as an annotation into a Bayesian framework which

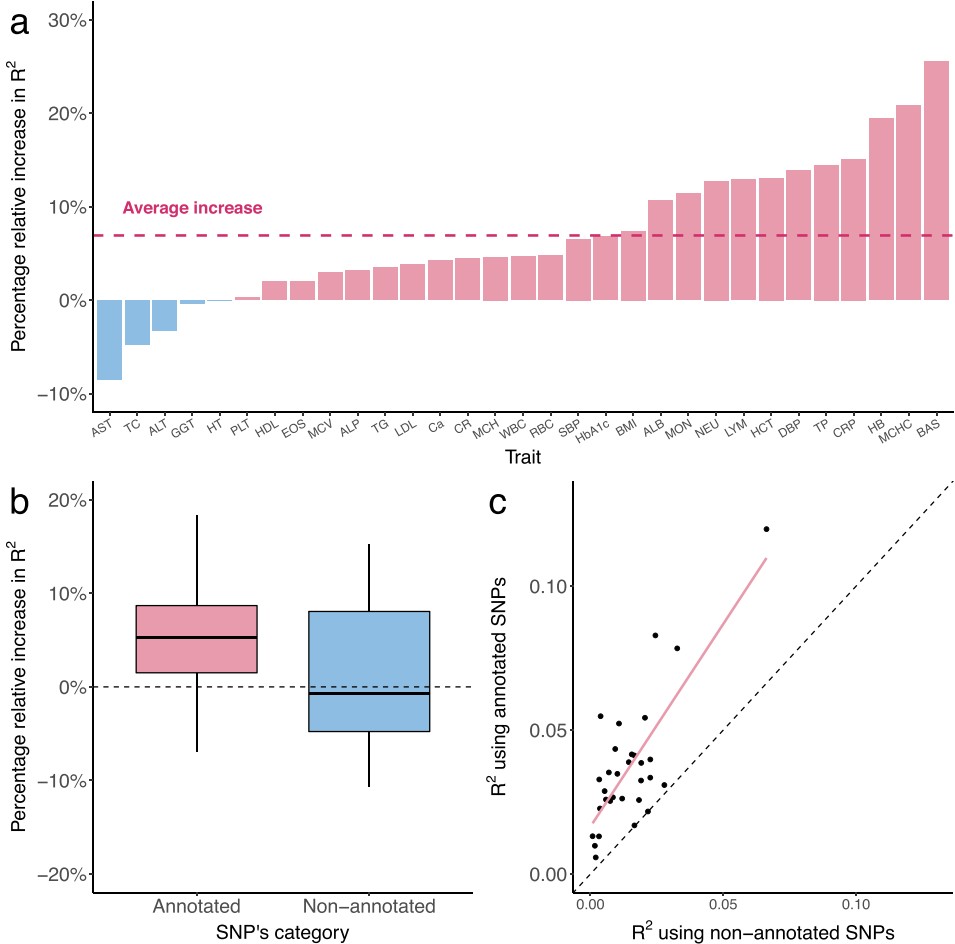

**Fig. 4 | Local genetic correlation annotation improves PRS prediction accuracy for 31 traits in East Asians. a** The percentage relative increase in $R^2$ for prediction accuracy of annotation-informed European PRS over PRS-CSx European PRS. A list of trait acronyms can be found in Supplementary Data 7. **b** The percentage relative increase in $R^2$ for prediction accuracy of annotation-informed over PRS-CSx European PRS using only annotated and non-annotated SNPs ($n = 31$ traits). In the boxplot, the center line, box limits and whiskers denote the median, upper and lower quartiles, and $1.5 \times$ interquartile range, respectively. **c** Comparison of $R^2$ between annotation-informed European PRS using only annotated and non-annotated SNPs. Each point represents a trait. X-axis is the $R^2$ for PRS based on non-annotated SNPs. Y-axis is the $R^2$ for PRS based on annotated SNPs.

amplifies correlated SNP effects between populations through annotation-dependent statistical shrinkage. It also combines multiple population-specific PRS to further improve prediction accuracy while using GWAS summary data alone as input. Applied to numerous GWAS traits, we demonstrated that local genetic correlations help pinpoint portable genetic effects and the annotation-informed PRS shows consistently and substantially improved performance across populations.

Our study presents several methodological innovations that will likely be generalizable and impactful. First, we introduced the concept of cross-population local genetic correlation and developed a scan statistic method to map correlated regions. Complementary to global genetic correlation, local genetic correlation refines the resolution in identifying shared genetic components between populations and provides critical insights into the genetic architecture of complex traits in diverse human populations. Second, we developed a new Bayesian framework that allows the integrative analysis of functional annotation data in multi-population PRS modeling. In this work, we showcased its effectiveness in cross-population risk prediction using an annotation derived from local genetic correlations. But we note that it is a general framework that can incorporate arbitrary sets of annotation data, such as the epigenetic annotations used in the PRS literature, in silico variant annotations based on machine learning exercises, or LD and allele

frequencies which have been shown to improve heritability estimation[21,23,33,55–57] (Supplementary Methods). It may also be applied to improve PRS portability across other non-ancestry-related demographic groups[58]. Finally, we introduced a strategy to combine multiple population-specific PRS into one improved score using summary statistics alone. This is innovative since fitting a regression model in an independent sample has long been considered the standard (and only) approach for combining multiple scores. This represents a significant advance in the field since obtaining additional individual-level samples that are independent from input GWAS can be a major challenge in practice. This is also generalizable since the same technique could be used to improve any PRS by creating an "omnibus" score over a number of methods, and the application is not limited to trans-ancestry risk prediction.

In addition to these methodological innovations, our local genetic correlation analysis identified many regions that are of biological interest. We have demonstrated that genomic regions identified by our approach show a substantial effect correlation on basophil count between two populations despite the low genetic correlation estimated from genome-wide data. More specifically, a region spanning 219 KB on chromosome 3 shows correlated effects between Europeans and East Asians for basophil count (Supplementary Fig. 25). Candidate gene *GATA2* at this locus encodes a zinc-finger transcription factor

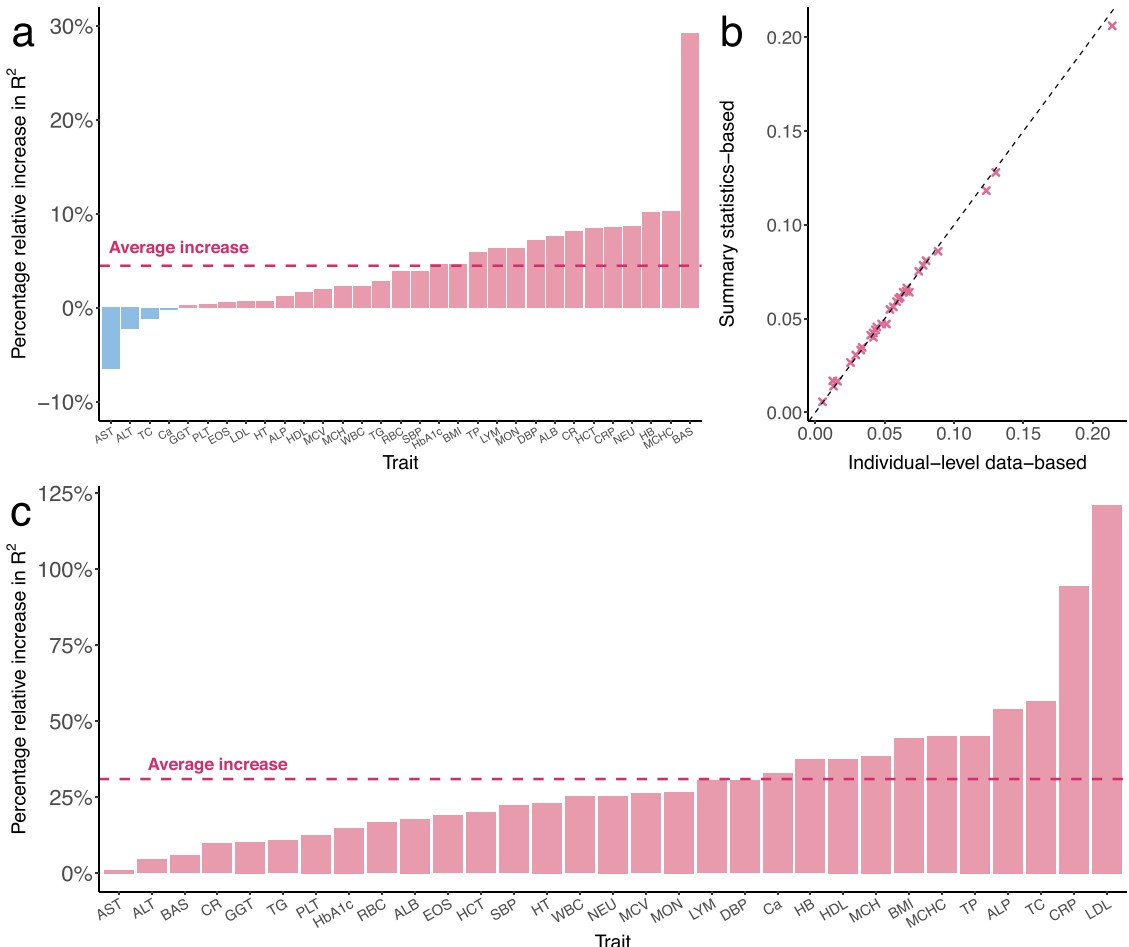

**Fig. 5 | Performance of X-Wing in combining population-specific PRS using GWAS summary statistics for 31 traits in East Asian samples. a** The percentage relative increase in $R^2$ of X-Wing PRS over PRS-CSx. The dashed line represents the average increase. A list of trait acronyms can be found in Supplementary Data 5. **b** Comparison of $R^2$ for linearly combined PRS with mixing weights obtained using GWAS summary statistics and individual-level data. The X-axis represents the $R^2$ using weights estimated from individual-level data, while the Y-axis shows the $R^2$ using summary statistics-based weights. The dashed line represents the diagonal line of y = x. **c** The percentage relative increase in $R^2$ of X-Wing PRS over PRS-CSx using GWAS summary statistics. PRS-CSx PRS is calculated based on European posterior mean effects. The dashed line represents the average increase.

which plays an essential role in proliferation, differentiation, and survival of hematopoietic cells[59]. In particular, expression of *GATA2*, coupled with CCAAT enhancer-binding protein α (C/EBPα) and transcription factor *STAT5*, directs the differentiation of granulocyte/monocyte progenitors (GMPs) into basophils[60,61]. Another correlated region for basophil count is a locus spanning 51 KB on chromosome 3 (Supplementary Fig. 26). Gene *IL5RA*, which encodes a subunit of a heterodimeric cytokine receptor that specifically binds to interleukin-5 (IL-5), lies 13 KB away from the identified region. Binding of the receptor to its ligand IL-5 is required for the biological activity of IL-5. Notably, IL-5 is a human basophilopoietin that promotes the formation and differentiation of human basophils[62,63]. Many other traits have interesting findings too. For example, a region spanning 48 KB on chromosome 1 is associated with C-reactive protein in two populations (Supplementary Fig. 27). The locus covers the gene *NLRP3*, which was identified as a risk gene associated with C-reactive protein levels in an independent GWAS[64]. *NLRP3* encodes a pyrin-like protein that constitutes the *NLRP3* inflammasome complex[65]. It was suggested that the *NALP3* inflammasome can activate nuclear factor-κB signaling[66] which affects C-reactive protein levels in Hep3B cells[64,67]. These results provide insights into the shared genetic basis of complex traits across ancestrally diverse populations. The local genetic correlation estimation procedure implemented in X-Wing may have broad applications in

future studies that involve joint modeling of multi-population GWAS associations.

Our study also has some limitations. First, although our method does not require any individual-level sample with both genotype and phenotype information, it remains crucial to have LD reference panels that match the input GWAS. We observed an improvement in PRS performance when applying our method to highly diverse samples such as the PAGE study, but it remains unclear how to best select LD references for multi-ancestry GWAS and admixed populations[68]. Second, we generally believe that statistical methods alone cannot fully solve the challenges in cross-population risk prediction[14,17]. It is an important future direction to apply state-of-the-art methods to the large and highly diverse GWAS conducted in global biobank cohorts[36], and carefully benchmark/combine various annotation data types and PRS training procedures. Third, although we have demonstrated an overall improved prediction accuracy over alternative methods across many traits, the relative improvement in $R^2$ reported for a single trait may be statistically imprecise (Supplementary Data 12) and should be interpreted with caution. Fourth, our simulations were carried out using HapGen2-simulated genotypes, which is known to have smaller fixation index ($F_{ST}$) than expected between two populations. Fifth, only categorical annotations were used for PRS construction in our analysis. It may be of interest to directly estimate local genetic correlation first,

and then incorporate the correlation values as a quantitative annotation to improve PRS.

Finally, the overall superior performance of X-Wing can be attributed to the incorporation of cross-population local genetic correlation and summary statistics-based PRS combination. Although we anticipate improved prediction accuracy after incorporating the local genetic correlation annotation, imprecise estimation of local genetic correlation may affect PRS performance when input GWAS have limited sample size. However, the summary statistics-based PRS combination strategy is robust in our analyses. In cases where there are concerns about the quality of local genetic correlation estimation, integrating summary statistics-based PRS combination into existing methods[16,23] should still be a strategy for consideration.

Taken together, X-Wing addresses major challenges in existing PRS methods, showcases multiple innovations in trans-ancestry GWAS modeling, and substantially improves the prediction accuracy of PRS in non-European populations. These methodological advances, in conjunction with the ever-growing GWAS sample size especially in non-European populations, give hope to broad and equitable applications of genomic precision medicine around the globe.

## Methods

### Quantifying local genetic correlations between ancestral populations

We extend the LOGODetect[38] framework to detect genomic regions showing local genetic correlations between two ancestral populations. Suppose the association z-scores for two populations are denoted as $\mathbf{z}_k = \frac{1}{\sqrt{N_k}}\mathbf{X}_k^\top \mathbf{Y}_k, k = 1, 2$. Here, $\mathbf{Y}_k$ is a $N_k$-dimensional vector of standardized phenotype values with mean 0 and variance 1, and $\mathbf{X}_k$ is the standardized genotype matrix of dimension $N_k \times M$ where $N_k$ is the GWAS sample size for population $k$. We define the scan statistic as

$$Q(R) = \frac{\sum_{i \in R} z_{1i} z_{2i}}{\left(\sum_{i \in R} \Sigma_{1,ii} * \Sigma_{2,ii}\right)^\theta} \quad (1)$$

where $R$ is the index set for SNPs in a genomic region, $\Sigma_k$ is the variance-covariance matrix of $\mathbf{z}_k$ and $\Sigma_{k,ii}$ denotes the i-th diagonal element of $\Sigma_k$. We note that the $\Sigma_k$ matrix can be estimated using $\Sigma_k = \frac{N_k h_k^2}{M} \widetilde{\mathbf{V}_k^2} + \left(1 - h_k^2\right)\mathbf{V}_k$. Here, $h_k^2$ is the trait heritability which can be estimated using GWAS summary statistics[25], $\mathbf{V}_k$ is the LD matrix which can be estimated using a reference panel, $\widetilde{\mathbf{V}_k^2} = \frac{N_k^{(ref)} - 1}{N_k^{(ref)} - 2}\mathbf{V}_k^2 - \frac{M}{N_k^{(ref)} - 2}\mathbf{V}_k$ is an unbiased estimator of the squared LD matrix, and $N_k^{(ref)}$ is the sample size of the LD reference panel. The numerator in the scan statistic is the inner product of association z-scores for two populations in a genomic region, which quantifies the correlation of SNP effect sizes in a genomic region. The denominator in the scan statistic adjusts for the effect of LD in two populations, where a tuning parameter $\theta$ controls the impact of LD. Technical details of the scan statistic and selection procedure for $\theta$ can be found in the Supplementary Methods.

To perform statistical inference, we use the maximal scan statistic over all possible genomic regions as the test statistic:

$$Q_{max} = \max_{|R| \le C} |Q(R)|, \quad (2)$$

where $C$ controls the upper bound of the region size (i.e., number of SNPs) and is pre-specified as 2000 in our analyses. Similar to local genetic correlation analysis in a single population[38], we draw 5000 Monte Carlo simulations of z-scores for each population to assess the null distribution of $Q_{max}$, and we apply the scanning procedure to identify significant genomic regions showing cross-population local

genetic correlations. Significant regions with a distance less than 100KB in-between are merged into a single segment.

### An annotation-dependent Bayesian horseshoe regression model for PRS

Next, we describe our Bayesian PRS framework with annotation-dependent statistical shrinkage. Consider an additive genetic model:

$$\mathbf{Y}_k = \mathbf{X}_k \boldsymbol{\beta}_k + \boldsymbol{\epsilon}_k, \boldsymbol{\epsilon}_k \sim MVN(\mathbf{0}, \sigma_k^2 \mathbf{I}_k), p(\sigma_k^2) \propto \sigma_k^{-2}, k = 1, 2, \ldots K, \quad (3)$$

where $\boldsymbol{\beta}_k$ is a $M$-dimensional vector of SNP effect sizes in population $k$, $\boldsymbol{\epsilon}_k$ is a vector of error terms with variance $\sigma_k^2$, to which we assign a non-informative Jeffreys prior[69]. $MVN$ denotes multivariate normal distribution, and $\mathbf{I}_k$ is an identity matrix.

We introduce an annotation-dependent shrinkage parameter, in addition to the global and local shrinkage parameters used in literature[16], to employ variable degrees of statistical shrinkage for SNPs in different annotation categories[42,43,45]. Here we only consider one annotation for simplicity, but our model allows incorporating multiple annotations (Supplementary Methods). Consider an annotation with $A$ categories, we assign an annotation-dependent horseshoe prior to $\beta_{jk}$:

$$\beta_{jk} \sim N\left(0, \frac{\sigma_k^2}{N_k}\phi\psi_j\lambda_{f(j),k}\right) j = 1, 2, \ldots M, k = 1, 2, \ldots K. \quad (4)$$

Here, $\beta_{jk}$ denotes the effect of SNP $j$ in population $k$, $\phi$ is the global shrinkage parameter shared across all $M$ SNPs and $K$ populations, $\psi_j$ represents the local shrinkage parameter for SNP $j$, $\lambda_{f(j),k}$ denotes the annotation-dependent shrinkage parameter for SNP $j$ in population $k$, $f : j \to a \in \{1, \ldots A\}$ is a function that maps the $j$-th SNP to its corresponding category $a$ in the annotation. The annotation-dependent shrinkage parameter is shared across SNPs that are in the same annotation category for a given population, but varies between populations to account for population-specific annotation.

Given this prior and marginal least squares estimates $\hat{\boldsymbol{\beta}}_k$ obtained from GWAS summary statistics, posterior mean effects in population $k$ is

$$E\left[\boldsymbol{\beta}_k | \hat{\boldsymbol{\beta}}_k\right] = \left(\mathbf{D}_k + \mathbf{S}_k^{-1}\right)\hat{\boldsymbol{\beta}}_k, \quad (5)$$

where $\mathbf{S}_k = diag\left\{\phi\psi_1\lambda_{f(1),k}, \phi\psi_2\lambda_{f(2),k}, \ldots, \phi\psi_M\lambda_{f(M),k}\right\}$ and $\mathbf{D}_k$ is the LD matrix for population $k$.

To provide an intuition of annotation-dependent statistical shrinkage, suppose all SNP are unlinked (i.e., no LD), then the LD matrix $D_k = I$ and the posterior mean effect for SNP $j$ in population $k$ is

$$E\left[\beta_{jk} | \hat{\beta}_{jk}\right] = \frac{1}{1 + \phi^{-1}\lambda_{f(j),k}^{-1}\psi_j^{-1}}\hat{\beta}_{jk} = \left(1 - \frac{1}{1 + \phi\lambda_{f(j),k}\psi_j}\right)\hat{\beta}_{jk} \quad (6)$$

Since SNPs in an important annotation explain more phenotypic variance ($\lambda_{f(j),k}$ tends to be big), the shrinkage factor $1 - \frac{1}{1 + \phi\lambda_{f(j),k}\psi_j}$ will be small if the j-th SNP is in an important annotation. Consequently, there is less statistical shrinkage on SNP effects in genomic regions marked by an important annotation.

To perform the full Bayesian model fitting, we assign half-Cauchy priors to the global, local, and annotation-dependent shrinkage parameters as follows:

$$\psi_j^{\frac{1}{2}} \sim C^+(1), \phi^{\frac{1}{2}} \sim C^+(1), \lambda_{a,k}^{\frac{1}{2}} \sim C^+(1) \ j = 1, 2, \ldots M, k = 1, 2, \ldots K, a = 1, 2, \ldots, A, \quad (7)$$

where $C^+(1)$ is the standard Cauchy distribution with the scale parameter equal to 1.

We employ a simple and efficient block Gibbs sampler to fit the PRS model using GWAS summary statistics and LD reference panel (Supplementary Methods)[70]. Following Ruan et al.[16], we recommend using $1000 \times K$ Markov Chain Monte Carlo (MCMC) iterations with the first $500 \times K$ iterations as burn-in. We use the full Bayesian approach as default, which does not require validation data to tune the model. An alternative strategy is to select the optimal global shrinkage parameter $\phi$ from $\{10^{-6}, 10^{-4}, 10^{-2}, 1\}$ that maximized the $R^2$ in the validation sample (Supplementary Methods)[16]. Our method outputs the posterior mean of population-specific SNP effects. PRS for the target cohort is calculated subsequently as the sum of allele counts weighted by posterior effect estimates.

### Incorporating local genetic correlation annotation in PRS

Below we explain how to incorporate annotations based on local genetic correlation in our PRS model. Without loss of generality, we assume population 1 is the target population. We break down our algorithm into three steps:

**Step1: Obtain annotation information through local genetic correlation analysis.** We perform local genetic correlation analysis between population 1 and population $k$ ($k = 2, \dots K$) to identify top $s$ regions with positive local genetic correlation. We denote the set of regions as $\Omega_k$ (e.g., when using UKB, BBJ, and PAGE as training GWAS, we ran local genetic correlation analysis between UKB and PAGE, as well as between BBJ and PAGE). We selected $s = 1000$ in our primary analysis and demonstrated that PRS performance is robust to the choice of $s$ (Supplementary Figs. 12, 13). We also used regions with both positive and negative local genetic correlation as annotation and demonstrated that the PRS performs better when only positive regions are used (Supplementary Fig. 28).

**Step2: Estimate posterior mean effects for all SNPs.** Our annotation-dependent shrinkage procedure is designed based on two key intuitions. First, we expect poor PRS portability when using GWAS from various ancestral populations (e.g., European and African) to predict trait values in a different target population (e.g., East Asian), Therefore, we want to amplify SNP effects that are more portable (i.e. correlated) between each non-target population and the target population. Second, we do not expect any portability issue when the GWAS population and the target population are the same (e.g., using an East Asian GWAS to build PRS for East Asian target samples). Thus, we do not employ any annotation-dependent shrinkage when estimating posterior SNP effects for the target population.

Specifically, when estimating posterior SNP effects for the target population, we let $\lambda_{f(j), k} = 1$ for all $j = 1, 2, \dots M$, $k = 1, \dots K$. When estimating the posterior SNP effects for the non-target population $k$ ($k = 2, \dots K$), we used $\lambda_{f(j),k} = \lambda_{1,k}$ if SNP $j$ is not annotated by $\Omega_k$, $\lambda_{f(j),k} = \lambda_{2,k}$ if SNP $j$ is annotated by $\Omega_k$, and $\lambda_{f(j),k'} = \lambda_{1,k'}$ for $k' = 1, \dots, k-1, k+1, \dots, K$. We provide an example for the case where $K = 3$ in the Supplementary Methods.

**Step3: Linearly combine multiple population-specific PRS.** Based on the posterior mean effects of population $k$ obtained in step2, we can calculate population-specific score $\mathbf{PRS}_k$. A common practice to combine these population-specific scores is to fit a regression model using the same phenotype $\mathbf{Y}^{(v)}$ and $K$ population-specific PRS in an independent validation dataset from the target population:

$$\mathbf{Y}^{(v)} \sim w_1 \mathbf{PRS}_1^{(v)} + w_2 \mathbf{PRS}_2^{(v)} + \dots + w_K \mathbf{PRS}_K^{(v)}. \tag{8}$$

Here, superscript $v$ highlights the fact that phenotypes and PRS in this regression exercise need to be obtained from a validation dataset that is different from any data used for GWAS and PRS modeling training. Instead of fitting a regression in independent samples, we

introduce a strategy to obtain the least squares estimates of regression weights (i.e. $\hat{w}_1, \dots \hat{w}_K$) using GWAS summary statistics. We introduce this approach in the next section. The final X-Wing PRS is then calculated as:

$$\mathbf{PRS}_{\mathrm{LC}} = \sum_{k=1}^{K} \hat{w}_k \mathbf{PRS}_k \tag{9}$$

### Combining multiple PRS with GWAS summary statistics

First, we briefly illustrate that we do not need any individual-level data from the validation sample, and summary statistics is sufficient for estimating the least squares estimator $\hat{\mathbf{w}}$ of PRS combination weights. Then, we provide detailed justifications on how to estimate $\hat{\mathbf{w}}$ using only input GWAS data instead of summary statistics from a validation sample. Suppose we have a validation dataset of $N^{(v)}$ individuals, $\hat{\mathbf{w}}$ can be estimated as follows:

$$\hat{\mathbf{w}} = \left[ \mathbf{PRS}^{(v)\mathrm{T}} \mathbf{PRS}^{(v)} \right]^{-1} \mathbf{PRS}^{(v)\mathrm{T}} \mathbf{Y}^{(v)}. \tag{10}$$

Here, $\mathbf{Y}^{(v)}$ is the phenotype vector and $\mathbf{PRS}^{(v)}$ is the $N^{(v)} \times K$ matrix of $K$ population-specific scores in this sample. Further, $\mathbf{PRS}^{(v)}$ can be denoted as $\mathbf{PRS}^{(v)} = \mathbf{X}^{(v)} \mathbf{b}$ where $\mathbf{X}^{(v)}$ is the $N_v \times M$ genotype matrix and $\mathbf{b}$ is the $M \times K$ matrix for SNP effects. For simplicity, we assume $\mathbf{Y}^{(v)}$ is centered, $\mathbf{X}^{(v)}$ is standardized, and $\mathbf{b}$ quantifies standardized SNP effects. We note that $\mathbf{PRS}^{(v)\mathrm{T}} \mathbf{PRS}^{(v)} / N^{(v)}$ quantifies the covariance of $K$ population-specific PRS which can be approximated by the sample covariance obtained from a reference panel (e.g., LD reference of the target population). Therefore, we have

$$
\begin{aligned}
\hat{\mathbf{w}} &= \left[ \mathbf{PRS}^{(v)\mathrm{T}} \mathbf{PRS}^{(v)} \right]^{-1} \mathbf{PRS}^{(v)\mathrm{T}} \mathbf{Y}^{(v)} \\
&= \left[ \mathbf{b}^{\mathrm{T}} \mathbf{X}^{(v)\mathrm{T}} \mathbf{X}^{(v)} \mathbf{b} \right]^{-1} \mathbf{b}^{\mathrm{T}} \mathbf{X}^{(v)\mathrm{T}} \mathbf{Y}^{(v)} \\
&= \left[ N^{(v)} \mathbf{b}^{\mathrm{T}} \frac{\mathbf{X}^{(v)\mathrm{T}} \mathbf{X}^{(v)}}{N^{(v)}} \mathbf{b} \right]^{-1} \mathbf{b}^{\mathrm{T}} \mathbf{X}^{(v)\mathrm{T}} \mathbf{Y}^{(v)} \\
&\approx \left[ N^{(v)} \mathbf{b}^{\mathrm{T}} \frac{\mathbf{X}^{(\mathrm{ref})\mathrm{T}} \mathbf{X}^{(\mathrm{ref})}}{N^{(\mathrm{ref})}} \mathbf{b} \right]^{-1} \mathbf{b}^{\mathrm{T}} \mathbf{X}^{(v)\mathrm{T}} \mathbf{Y}^{(v)} \\
&= \frac{N^{(ref)}}{N^{(v)}} \left[ \mathbf{PRS}^{(\mathrm{ref})\mathrm{T}} \mathbf{PRS}^{(\mathrm{ref})} \right]^{-1} \mathbf{b}^{\mathrm{T}} \mathbf{X}^{(v)\mathrm{T}} \mathbf{Y}^{(v)}
\end{aligned}
\tag{11}
$$

where $\mathbf{X}^{(v)\mathrm{T}} \mathbf{Y}^{(v)}$ can be obtained from the summary statistics of the validation sample (Supplementary Methods) and $\mathbf{b}$ is obtained from the PRS training procedure. $N^{(ref)}$ and $\mathbf{PRS}^{(\mathrm{ref})}$ denote the sample size and PRS matrix in the reference panel. Taken together, Eq. (14) shows that LD reference and summary statistics from a validation sample can be used to estimate $\hat{\mathbf{w}}$. However, summary statistics from a validation cohort are still difficult to obtain in practice, and it is tempting to replace it with the input GWAS used for PRS training. But this is not feasible since it is a textbook example of overfitting. This motivates us to use repeated learning (or a similar cross-validation approach; see Supplementary Methods)[71,72] to estimate $\hat{\mathbf{w}}$.

Typically, repeated learning (or cross-validation) requires individual-level genotype and phenotype data since it involves sample splitting. Generalizing the technique in our recent work[8] and its extension handle the LD[33], we introduce a summary statistics-based repeated learning strategy, which mimics the individual-level repeated learning but does not need individual-level GWAS data (Supplementary Methods). This approach has three main steps which we describe below. Since this approach does not involve a separate validation sample, we will perform analysis using input GWAS from the target population (e.g., BBJ GWAS when East Asian is the target population), the sample size of which is typically sufficiently large to ensure the

performance of repeated learning. Without loss of generality, we denote $k = 1$ for this (target) population.

**Step1: Subsample GWAS summary statistics from training and validation sets.** Suppose we divide the full GWAS sample ($\mathbf{X}_1$, $\mathbf{Y}_1$) into a training set ($\mathbf{X}_1^{(tr)}$, $\mathbf{Y}_1^{(tr)}$) with $N_1 - N_1^{(v)}$ individuals, and a validation set ($\mathbf{X}_1^{(v)}$, $\mathbf{Y}_1^{(v)}$) with $N_1^{(v)}$ individuals. Given the association z-scores ($\frac{\mathbf{X}_1^T \mathbf{Y}_1}{\sqrt{N_1}}$) from GWAS summary statistics and genotype data from the reference panel, association summary statistics based on training and validation sets can be sampled as:

$$\frac{\mathbf{X}_1^{(tr)T}\mathbf{Y}_1^{(tr)}}{N_1 - N_1^{(v)}} = \frac{\mathbf{X}_1^T\mathbf{Y}_1}{N_1} + \left(\frac{N_1^{(v)}}{N_1(N_1 - N_1^{(v)})}\right)^{\frac{1}{2}}\frac{\mathbf{X}^{(ref)T}}{\sqrt{N^{(ref)}}}\mathbf{g}$$
$$\frac{\mathbf{X}_1^{(v)T}\mathbf{Y}_1^{(v)}}{N_1^{(v)}} = \frac{\mathbf{X}_1^T\mathbf{Y}_1 - \mathbf{X}_1^{(tr)T}\mathbf{Y}_1^{(tr)}}{N_1^{(v)}}, \tag{12}$$

where $\mathbf{X}^{(ref)}$ is a $N^{(ref)} \times M$ standardized genotype matrix from the reference panel for the target population, $N^{(ref)}$ is the sample size of the reference panel, $\mathbf{g}$ is a $N^{(ref)}$-dimensional vector with elements drawn from a standard normal distribution (Supplementary Methods).

**Step2: PRS model training.** We train our PRS model using the training summary statistics subsampled for the target population in step1 and full GWAS summary statistics (without subsampling) for other populations. The output of PRS training is a $M \times K$ matrix $\mathbf{b}$ with the $k$-th column showing standardized SNP effects for population $k$ (Supplementary Methods).

**Step3: Estimate the linear combination weights.** We then estimate PRS weights by

$$\hat{\mathbf{w}} \approx \frac{N^{(ref)}}{N_1^{(v)}}\left[\mathbf{PRS}^{(ref)T}\mathbf{PRS}^{(ref)}\right]^{-1}\mathbf{b}^T\mathbf{X}_1^{(v)T}\mathbf{Y}_1^{(v)}, \tag{13}$$

where $\mathbf{PRS}^{(ref)} = \mathbf{X}^{(ref)}\mathbf{b}$ denotes the $N^{(ref)} \times K$ PRS matrix calculated in the reference panel, $\mathbf{X}_1^{(v)T}\mathbf{Y}_1^{(v)}$ is the subsampled validation summary statistics. We note that when we calculate $\hat{\mathbf{w}}$ using PRS matrix in the reference panel, essentially only LD matrix is used: $\mathbf{PRS}^{(v)T}\mathbf{PRS}^{(v)} = \mathbf{b}^T\mathbf{X}^{(v)T}\mathbf{X}^{(v)}\mathbf{b} \approx \frac{N_1^{(v)}}{N^{(ref)}} \times \mathbf{b}^T\mathbf{X}^{(ref)T}\mathbf{X}^{(ref)}\mathbf{b} = \frac{N_1^{(v)}}{N^{(ref)}} \times \mathbf{PRS}^{(ref)T}\mathbf{PRS}^{(ref)}$, where $\frac{\mathbf{X}^{(ref)T}\mathbf{X}^{(ref)}}{N^{(ref)}}$ is the LD matrix. We choose to calculate $\hat{\mathbf{w}}$ using PRS matrix to reduce computational complexity compared to directly using LD matrix, but one can still estimate $\hat{\mathbf{w}}$ using only LD matrix in the reference panel (Supplementary Methods). In practice, we force any negative estimates $\hat{w}_k$ to be 0 and center PRS in the reference panel. We also normalize PRS weights by $\tilde{\mathbf{w}} = \frac{\hat{w}}{\sum_{k=1}^{K}\hat{w}_k}$.

At last, we perform *P*-fold repeated learning. The final linear combination weights $\hat{\mathbf{w}}_{final}$ is the average of the normalized mixing weights across *P* times:

$$\hat{\mathbf{w}}_{final} = \frac{\sum_{p=1}^{P}\tilde{\mathbf{w}}_p}{P}, \tag{14}$$

where $\tilde{\mathbf{w}}_p$ represents the normalized weights in *p*-th fold. To avoid overfitting, we used distinct reference panels from the target population for GWAS summary statistics subsampling, PRS model training, and estimating weights for PRS combination. We provide the equally divided reference panels from 1000G phase 3 data for Europeans, East Asians, Africans, Central/South Asians, and admixed Americans to the users. We also present the extensions of our approach to handle tuning parameters in PRS model, negative mixing weights

from least squares, and multicollinearity between PRS in Supplementary Methods.

## Simulations

We used HAPGEN2[73] to simulate genotypes for 50,000 individuals of European and East Asian ancestry respectively from population-matched 1000 Genomes Project data. We only included SNPs with MAF greater than 5% on chromosome 22. After removing strand-ambiguous variants, 55,000 SNPs remained in the dataset and were used for subsequent analysis.

First, we carried out simulations to assess the type I error rates of two methods (i.e., X-Wing and PESCA). We generated the effect size of each SNP for two populations independently (i.e., under the null) following an infinitesimal model, where the per-SNP heritability was fixed as a constant. Trait heritability for two populations were set to be the same and varied between 0.001 and 0.01. We also compared two methods in three additional model settings: heritability enrichment model, LDAK model[48] (SNP heritability is dependent on LD and MAF), and binary trait scenario. In the heritability enrichment model, 30% of heritability was attributed to 1000 randomly selected SNPs and 70% of heritability to the remaining SNPs. LDAK model assumes that the effect size of the *j*-th SNP follows the normal distribution N(0,$h_j^2$) and the per-SNP heritability $h_j^2$ is proportional to $\left[f_j^*\left(1 - f_j\right)\right]^{0.75}*u_j$, where $f_j$ is MAF and $u_j$ is LDAK weight computed by the LDAK software. In the binary trait scenario, we first simulated the continuous liability following the same infinitesimal model as described above, then assigned the samples with top 50% liability as cases and others as controls. We repeated each simulation setting 100 times. Type I error rate was defined as the proportion of simulation repeats in which correlated regions (for X-Wing) and causal SNPs shared by two populations (for PESCA) were identified.

Next, we compared the statistical power of X-Wing and PESCA under the heritability enrichment model. We randomly selected a genome segment on chromosome 22 spanning 1000 SNPs as the correlated signal region. We attributed 30% trait heritability to the signal region. We jointly simulated SNP effect sizes in the correlated signal region for two populations with a correlation set as 0.9, and then simulated effect sizes of the rest of the genome independently between populations. Trait heritability for two populations were set to be the same and varied between 0.001 and 0.01. We also investigated the LDAK model and the binary trait model. Each simulation setting was repeated 100 times. Statistical power was defined as the proportion of simulation repeats in which at least one identified region (for X-Wing) and one shared causal SNP (for PESCA) overlapped with the true signal region. We also performed simulations across the whole genome. We simulated genotypes for 50,000 individuals and 831,636 HapMap3 SNPs using the HapGen2 software. We simulated two independent traits for two populations under the infinitesimal model and assessed the type-I errors for the two methods. To compared statistical power under the heritability enrichment model, we randomly selected 50 genome segments, each spanning 1000 SNPs as the correlated signal regions. 30% trait heritability was attributed to the signal regions and 70% was attributed to the rest of the genome. Correlation of SNP effect sizes in the correlated signal regions was set as 0.9. We further performed simulation to compare the predictive accuracy (measured by $R^2$) of X-Wing PRS with the existing methods PRS-CSx and XPASS (Supplementary Methods).

## Analysis of GWAS data from UKB, BBJ, and PAGE study

We evaluated the prediction accuracy of X-Wing PRS using 31 traits in East Asians and 13 traits in admixed Americans and Africans. European and East Asian GWAS summary statistics were obtained from UKB and BBJ (see Data availability). Trans-ancestry GWAS summary statistics for

13 traits were obtained from the PAGE study[74] (Supplementary Data 5). East Asian and admixed American target samples in UKB were identified based on the Pan-UKB population assignment[75]. We removed samples already included in the UKB European GWAS. We also used KING[76] to infer sample relatedness, and only kept individuals without any relatives at the third-degree or higher. We further excluded individuals with conflicting genetically-inferred and self-reported sex. The final East Asian, admixed American, and African target sample consist of 2683, 749, and 6490 individuals, respectively. We calculated PRS for these samples using the imputed genotype data provided by UKB but restricted to the autosomal SNPs with info score > 0.9, MAF > 0.01, missing rate $\leq$ 0.01, and Hardy Weinberg equilibrium test $p$-value $\geq$ 1.0e-6.

We applied X-Wing to obtain the annotations based on pairwise local genetic correlation between European, East Asian, and admixed American population using UKB, BBJ, and PAGE GWAS summary statistics. We annotated SNPs in the top 500, 1000, 1500 correlated regions and excluded regions with negative correlations. We then incorporated the annotation into our PRS model, using 1000 G phase3 data provided in Ruan et al.[16] as LD reference panel and independent LD block provided by LDetect[77] for block Gibbs sampler. When the target population is East Asian, we used UKB and BBJ GWAS as training data and European and East Asian LD reference panel. For the admixed American and African target population, we used UKB, BBJ, and PAGE GWAS as training data and European, East Asian, and admixed American LD reference panel, since PAGE GWAS consists primarily of Hispanic/Latino[16]. We randomly and evenly split the target cohort into a validation dataset to linearly combine population-specific PRS and used the remaining samples as the test dataset to evaluate PRS performance. When the PRS model involves model-tuning, the validation dataset is also used to select tuning parameters. We used partial $R^2$ averaged across 100 random splits to benchmark the predictive accuracy of different methods, adjusting for age, sex, $age^2$, age × sex, $age^2$ × sex, and the top 20 genetic principal components. We used the percentage increase in partial $R^2$ for X-Wing over other methods and reported the p-value from two-sided Wilcoxon signed-rank test to compare their performance. X-Wing uses local genetic correlation annotations based on genome-wide imputed SNPs in primary analysis but shows almost identical results using annotations based on HapMap3 SNPs (Supplementary Fig. 29). When the target population is Africans, we further replaced the admixed American LD reference panel with European or Africans LD reference panel and found that using admixed American LD reference yields better predictive performance over alternatives (Supplementary Fig. 30).

We implemented 4-fold repeated learning to estimate the PRS combination weights using GWAS summary statistics and our equally divided 1000G reference panel[8,78]. In each fold, we first subsampled East Asian (or admixed American) summary statistics for 75% BBJ (or PAGE study) samples as the training and the remaining 25% as the validation set. We applied X-Wing using the UKB and subsampled 75% BBJ training data (or UKB, BBJ, and 75% simulated PAGE summary statistics) to obtain the posterior mean effects for each population. We then used these posterior mean effects to calculate PRS in the 1000G dataset for East Asian (or admixed American) samples and estimated the linear combination weights. We calculated the average weight values over four repeats, used these weights to combine population-specific PRS, and compared its prediction accuracy with the combined PRS based on individual-level data in the same target population. The weights selected from our repeated learning procedure for 29/31 traits in East Asians falls into the 95% confidence interval of the weights estimated in an independent sample (Supplementary Fig. 31). X-Wing uses 4-fold repeated learning in primary analysis but shows almost identical results using 10-fold repeated learning (Supplementary Fig. 32). In our software implementation, we allow the users to specify the number of folds in repeated learning.

Implementation details of XPASS, PESCA, PolyFun-pred, PolyPred+ and PRS-CSx are described in the Supplementary Methods.

### Reporting summary
Further information on research design is available in the Nature Portfolio Reporting Summary linked to this article.

## Data availability
This study made use of publicly available datasets. This research has been conducted using the UK Biobank Resource under Application Number 42148. Data from the UK Biobank are available by application to all bona fide researchers in the public interest at https://www.ukbiobank.ac.uk/enable-your-research/apply-for-access. Phase 3 data of the 1000 Genomes Project are publicly available at ftp://ftp.1000genomes.ebi.ac.uk/vol1/ftp/release/20130502/; Pan UK Biobank data are publicly available at: https://pan.ukbb.broadinstitute.org; UKB GWAS summary statistics data are publicly available at: http://www.nealelab.is/uk-biobank; BBJ GWAS summary statistics data are publicly available at: http://jenger.riken.jp/en/result; PAGE study GWAS summary statistics data are publicly available at: https://www.ebi.ac.uk/gwas/publications/31217584; PolyFun-pred PRS coefficients data are publicly available at: http://data.broadinstitute.org/alkesgroup/polypred_results.; All data generated during this study are included in this published article and its supplementary information files. X-Wing posterior SNP effect size estimates in this work are publicly available at https://github.com/qlu-lab/X-Wing.

## Code availability
X-Wing software is freely available at https://github.com/qlu-lab/X-Wing;

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

## Acknowledgements

We thank Drs. Lauren Schmitz and Jason Fletcher for helpful discussions. Q.L. and J.M. are supported by the University of Wisconsin-Madison Office of the Chancellor and the Vice Chancellor for Research and Graduate Education with funding from the Wisconsin Alumni Research Foundation (WARF). L.H. acknowledges research support from the National Natural Science Foundation of China (Grant No. 12071243).

## Author contributions

J.M., H.G., L.H., and Q.L. conceived and designed the study. J.M. developed the statistical frameworks for incorporating annotation data into multi-ancestry PRS modeling and combining multiple PRS with GWAS summary data. H.G. developed the method for quantifying the local genetic correlation between distinct populations. J.M. and H.G. performed statistical analyses. G.S. assisted in preparing GWAS summary statistics. Z.Z. assisted in implementing summary statistics-based repeated learning. L.H. and Q.L. advised on statistical and genetic issues. J.M., H.G., L.H., and Q.L. wrote the manuscript. All authors contributed in manuscript editing and approved the manuscript.

## Competing interests

The authors declare no competing interests.
