## [Peer Review File · Nature Communications]

Quantifying portable genetic effects and improving cross-ancestry genetic prediction with GWAS summary statisticsREVIEWER COMMENTS

Reviewer #1 (Remarks to the Author):

In this paper, the authors describe a novel method, X-Wing, for the genetic prediction of ancestrally diverse populations. The proposed method X-Wing builds upon the Bayesian and scan statistics framework and aims to enhance the predictive accuracy using the information of cross-population local genetic correlation. The X-Wing method addresses an important issue of the cross-ancestry polygenic risk score calculation using summary-level data (summary statistics and LD-reference) only. The authors demonstrate the advantage of the proposed method using simulations and GWAS data in UK biobank, Biobank Japan, and PAGE study. The text overall is very clearly written.

However, I do have some major concerns that I would like the authors to address.

(1) The simulation times of type I error control for X-Wing are only 100 times, which is not enough to show the accuracy at the 0.05 level. Therefore, I recommend the reviewer to do at least the type I error simulation at least 1,000 times.

(2) The authors controlled the FDR at a 0.05 level for selecting the regions with significant cross-population local genetic correlations. The author needs to discuss more on the selection on this threshold, as the p-value threshold for SNP selection varies from trait to trait in a single population PRS. Is 0.05 the optimal threshold, or is the procedure robust to the threshold? A sensitivity analysis of this significant threshold would be helpful here.

(3) This reviewer agrees with the authors that newly implemented approaches are useful, and have advantages over existing methods in real data analysis. Still, it was not clear how they were actually beneficial. Therefore, this reviewer recommends conducting additional power simulations to compare X-Wing with the existing methods PRS-CSx and XPASS.

Minor comments

(1) Recent studies show that incorporating functional annotations represent different biological functionalities increase the association analysis power¹. This reviewer recommended the reviewer discuss the ability of X-Wing to incorporate multiple functional annotations of different biological aspects to further improve the accuracy, not only the epigenetics scores.

Reference

1. Li, X. et al. Dynamic incorporation of multiple in silico functional annotations empowers rare variant association analysis of large whole-genome sequencing studies at scale. *Nature genetics* 52, 969-983 (2020).

Reviewer #2 (Remarks to the Author):

In this paper, the Miao et al introduced the X-wing method, a novel PRS algorithm for cross-ancestry analyses. While X-wing only provide a marginal improvement over PRS-CSx, the authors introduce a procedure which allow parameter optimization without requiring additional genotype data. I do have a number of questions:

1. Judging from the equations, the main difference between X-wing and PRS-CSx is the addition of annotation-dependent shrinkage parameter, correct?
2. One of my main concern is that the authors only simulated using chromosome 22, which means the simulated heritability will be disproportionately large compared to realistic scenarios. In addition, based on my experience, when only simulating 1 chromosome, especially if it is on chr22, the sample generated by HapGen2 will tends to be highly related. A more realistic simulation should be across all the whole genome.
3. Just a note, when simulating using HapGen2, the F_{st} between the two subpopulation will tends to be much smaller than what was expected. E.g. simulated AFR vs EUR samples might only have a F_{st} of 0.006 instead of >1 , which might worth pointing out in the limitation
4. In this paper, the authors mainly focuses on East Asian and European samples. However, when compared to African samples, the transferability of European based PRS to East Asian are usually easier. It would be worthwhile for the authors to also consider applying X-wing in the AFR samples similar to the PRS-CSx paper
5. Simulations were only done to compare the PESCA and X-wing, given the main purpose of X-wing, a more interesting simulation might be to compare the performance between X-wing and PRS-CSx to investigate in what scenarios PRS-CSx performs outperform X-wing and vis versa.
6. For the cross validation procedure, my understanding is that the author split the target samples (which can be from, say East Asian) in half, use half of those sample for GWAS, and then use the calculated summary statistics and the summary statistics from a reference cohort (e.g. from European) to calculate a PRS in the 1000 Genome (with matched population, eg. calculate EUR PRS in EUR 1000G, calculate X PRS in X 1000G where X is the target population). Finally, the weight were estimated using the genotype and phenotypes of the remaining target samples (the validation cohort), correct? In that case, what is the sample size limitation for this method to work? I can imagine that if the target cohort size is small e.g. < 1000 , which is usually the case for under-representative populations, then the CV procedure might be highly unstable?

7. Follow up on that, in the paper, the authors only perform 4 fold cross validation, which seems like a relatively small number. Would that be enough? How does the number of fold affects the stability of the algorithm? I'd imagine that will be a function of the sample size (e.g. number of cases) and the number of folds?

8. My other concern regarding this procedure is that the reference population (e.g. the 1000 Genome) tends to ascertain for healthy samples. Would that lead to bias to your algorithm? How similar will the weight selected from this CV procedure compared to if we have an independent sample for optimization?

9. Why does the authors exclude the negatively correlated regions? Was there an implicit assumption that the direction of effect should be consistent across populations? Would relaxing this assumption negatively impact the performance of X-wing?

10. Does the chose of upper bound of region size, and/or the merge distance, and/ or the s parameter affects X-wing's performance?

Minor comments:

1. When stating "gain in predictive R2" should specify that these are "relative gain"
2. for region size, is that the number of variants or size of a region (e.g. 2000 variants or 2000 bp?)
3. Given the addition complexity, and considering that PRS-CSx isn't known for its speed, how computationally intensive is X-wing?

Reviewer #3 (Remarks to the Author):

Miao and colleagues have developed a method called X-Wing with the aim to improve PRS transferability across ancestries. This is an important research question to achieve equitable benefits and uses of PRS in global populations. X-Wing uses information from between-ancestry local genetic correlation as functional annotations to estimate population-specific posterior mean effects. The authors ran both simulations and real data analyses to explore the reliability of identifying genomic regions with significant local genetic correlation. UKB, BBJ and PAGE datasets on various complex traits were further used to construct PRS using X-Wing as compared to other methods (PRS-CSx, PolyFun-Pred, XPASS). Regarding the method, a lot of extensions have been done on integrating the authors' previous work. The manuscript is overall well organized (with a well-documented github page). I have some comments below that hopefully improve the manuscript.

Major comments:

1) The strategy to detect regions showing local genetic correlation is an extension to the authors' previous work LOGODetect. But it is unclear how the method can be applied for more than two ancestral populations when the authors using UKB, BBJ and PAGE as the training datasets. Also, would high LD regions (such as MHC region) affect this and following analyses?

2) For the annotation-dependent Bayesian horseshoe regression model:

a. Categorical annotations were used in the study. As the region identified, it would be possible and reasonable to calculate the values of local genetic correlation. I was wondering whether using the specific values would further contribute to the PRS performance.

b. Marginal effects were assumed in the model. As most current GWAS (such as BBJ used in this study) are performed with mixed models, explorations or comments on how this will affect the model performance would be helpful.

c. I was not convinced by the rationale that annotation-dependent shrinkage was not applied for GWAS with same ancestry as target populations because there is no loss of portability. Would implementing annotations further improve the PRS performance for target population when using target-ancestry specific effects?

3) For linearly combine multiple PRS:

a. Related to previous marginal effects assumption, most GWAS are performed accounting for sample relatedness. Therefore, I think it will not be reasonable to apply splitting for such GWAS, although the results show the accuracy was not affected. The authors should account for this or justify its use in such scenarios.

b. I am confused about how the use of LD reference panel would result in overfitting. The authors divide the small 1000G cohorts to even smaller datasets, which overall could reduce the precision of LD approximation. Also, the authors mentioned in the discussion that the mismatch between LD reference panel and discovery GWAS could impact the PRS performance, especially for multi-ancestry GWAS such as PAGE. Descriptions about ancestry proportions of PAGE would be helpful, also please add what LD reference was used for PAGE. However, the impact of such LD mismatch was not explored here as 1000G is also an external LD reference panel. As X-Wing is dependent on LD reference panels, I think further exploration about the impact of them would be important to show the results won't be largely affected by them.

4) Regarding the accuracy comparisons among different methods, they were not intuitive and informative, specifically:

a. The accuracy is mainly reported based on relative improvement without standard errors, CIs or p-value estimates, thus it is hard to conclude whether the improvement is significant here. For example, the 4.1% increase in R^2 compared to PRS-CSx seems to be only a marginal increase based on the number. Please add such statistics.

b. The authors report the accuracy using EUR-based PRS and linearly combined multi-ancestry PRS using X-Wing in two sections, respectively. It would be helpful to see the direct comparisons between the two

PRS. If the linear combination of PRS did not significantly improve the accuracy, then it might suggest we can directly use the population specific PRS without combine them; if it increased, then comments on this would be helpful.

c. Directly comparing the accuracy of different methods in the same figure would be more intuitive and informative. Specifically, for PolyFun-Pred, the extensions of summary-level based PolyPred by integrating multiple populations/predictors and a few other methods with available coefficients would be worth to be compared.

d. The results show the improvement of X-Wing is largely dependent on trait-specific genetic architecture but without specific comments about this (except for BAS). Further, the traits here are all quantitative traits, thus it would be helpful to show a few disease traits.

e. Not clear about the SNPs included: Only HapMap3 SNPs used? Or genome-wide imputed variants? How can SNP density impact the results (especially the local genetic correlation)?

5) Extensions of analyses to African populations would be very helpful as PRS has the lowest portability in them, especially when PAGE was added in the discovery GWAS. But here AMR with small sample sizes from UKB was focused.

Minor comments:

1) In the introduction, “the third approach” or the third type of approaches?

2) Figure 2: please add descriptions for power and genetic covariance.

3) Figure 3C: please comment on why combining discovery and replication stage results in less identified regions? How could the number of regions affect by various factors?

4) Figure 5, why PRS-CSx is only based on EUR-based PRS when comparing to linearly combined PRS using X-Wing?

5) The results show that most genomic regions have positive genetic correlation, which are interesting and convincing. They are consistent with some previous studies that common causal variants are largely shared between-ancestries. The authors noted a few regions with negative correlation and mentioned those regions were removed in the analyses in the methods. Do the authors have explanations for such negative correlations?

6) The descriptions about inputs required to estimate $w_{\hat{}}$ is not consistent, i.e., whether summary statistics from validation cohorts are needed or not. (“Then, we provide detailed justifications on how to estimate...”, “Taken together, ... we only need the LD reference and summary statistics from a validation sample.”)

7) In Figure S9 and S10, the improvement of X-Wing over PRS-CSx/XPASS seemed to vary with the number of regions selected. But the authors mentioned that “We selected $s = 1000$ in our primary analysis and demonstrated that PRS performance is robust to the choice of s ”, please use the same y-axis scales to present and run statistical test to see whether they are truly insignificant different.

8) The computation cost and requirements (e.g., whether can be run paralleled) is missing for X-Wing as compared to other methods, which would be helpful for future users.

9) How global genetic correlation was calculated here?

10) Are the individuals in HapGen2 all unrelated?

We thank the reviewers for their constructive comments. We have added text to clarify some technical details and performed additional simulations and analyses in our revised manuscript. Overall, we have made the following major updates to the manuscript compared to our initial submission:

1. We have added an analysis to benchmark PRS methods using 13 complex traits in Africans. We have also included PolyPred+ as a method for comparison in our analysis of East Asian samples and expanded the analysis to include a binary disease trait.
2. We have quantified the statistical evidence for the improvement of X-Wing prediction accuracy over existing PRS methods. We report P-values from the two-sided Wilcoxon signed-rank test.
3. We have substantially expanded our sensitivity analyses. In addition to varying the number of top annotation regions, we also assessed the impact of excluding the MHC region, using only Hapmap3 SNPs in creating annotations, using both positively and negatively correlated regions as annotations, using different upper bound of region size and merging distance, changing the LD reference panel for PAGE study, and varying the number of folds in repeated learning, to demonstrate the robust performance of X-Wing.
4. We have also expanded our simulation analysis by increasing the number of simulation repeats, extending the simulation to whole genome data, and adding new simulations focusing on PRS performance.
5. We have reported the runtime and memory usage of X-Wing.

We provide a point-to-point response (colored in blue) below. Changes to the text are highlighted in the revised manuscript.

Reviewer #1 (Remarks to the Author):

In this paper, the authors describe a novel method, X-Wing, for the genetic prediction of ancestrally diverse populations. The proposed method X-Wing builds upon the Bayesian and scan statistics framework and aims to enhance the predictive accuracy using the information of cross-population local genetic correlation. The X-Wing method addresses an important issue of the cross-ancestry polygenic risk score calculation using summary-level data (summary statistics and LD-reference) only. The authors demonstrate the advantage of the proposed method using simulations and GWAS data in UK biobank, Biobank Japan, and PAGE study. The text overall is very clearly written. However, I do have some major concerns that I would like the authors to address.

(1) The simulation times of type I error control for X-Wing are only 100 times, which is not enough to show the accuracy at the 0.05 level. Therefore, I recommend the reviewer to do at least the type I error simulation at least 1,000 times.

Response: Thank you for the constructive comment. Following your suggestion, we have added simulations to assess type-I errors under 1000 repeats. Both X-Wing and PESCA showed well-controlled type-I error rates under a variety of data generating models (**Supplementary Tables 1-4**).

Supplementary Table 1 in the revised manuscript. Type I error rates under the infinitesimal model.

Trait heritability	X-Wing	PESCA
0.001	0.029	0.002
0.004	0.033	0.008
0.007	0.027	0.035
0.01	0.047	0.034

Supplementary Table 2 in the revised manuscript. Type I error rates under the heritability enrichment model.

Trait heritability	X-Wing	PESCA
0.001	0.03	0.002
0.004	0.042	0.02
0.007	0.049	0.031
0.01	0.044	0.044

Supplementary Table 3 in the revised manuscript. Type I error rates under the LDAK model.

Trait heritability	X-Wing	PESCA
0.001	0.019	0.003
0.004	0.033	0.007
0.007	0.038	0.011
0.01	0.06	0.019

Supplementary Table 4 in the revised manuscript. Type I error rates for binary traits.

Trait heritability	X-Wing	PESCA
0.001	0.028	0.004
0.004	0.027	0.027
0.007	0.028	0.032
0.01	0.029	0.034

(2) The authors controlled the FDR at a 0.05 level for selecting the regions with significant cross-population local genetic correlations. The author needs to discuss more on the selection on this threshold, as the p-value threshold for SNP selection varies from trait to trait in a single population PRS. Is 0.05 the optimal threshold, or is the procedure robust to the threshold? A sensitivity analysis of this significant threshold would be helpful here.

Response: We appreciate this comment. In our paper, we used an FDR cutoff of 0.05 to claim statistically significant regions with local genetic correlations between populations. However, to improve risk prediction, we employed the top 1000 regions with positive local genetic correlation as an informative annotation to construct PRS. This is implemented in a way that users of X-Wing can choose their own cutoff or switch the annotation completely. In the revised manuscript, we have also performed sensitivity analysis on this choice. We obtained highly consistent prediction results using different numbers of top regions with local genetic correlations (**Supplementary Figure 12**).

Supplementary Figure 12. Comparison of the prediction accuracy between X-Wing and PRS-CSx PRS for 31 traits in East Asians with varying numbers of top regions. Panel **a**, **b**, and **c** show the percentage increase in R^2 of X-Wing European PRS over PRS-CSx using 500, 1000, and 1500 positive regions as annotation. Panel **d**, **e**, and **f** are the results for linearly combined PRS. The dashed line represents the average increase.

(3) This reviewer agrees with the authors that newly implemented approaches are useful, and have advantages over existing methods in real data analysis. Still, it was not clear how they were actually beneficial. Therefore, this reviewer recommends conducting additional power simulations to compare X-Wing with the existing methods PRS-CSx and XPASS.

Response: Thanks for the suggestion. We have added new simulations to compare the predictive accuracy (measured by R^2) of X-Wing with two existing methods, PRS-CSx and XPASS. We used HapGen2-simulated genotypes for Europeans and East Asians released in Zhang et al.¹, which contains 956,041 Hapmap3 SNPs after quality control. We randomly selected 50 genomic segments, each spanning 1,000 SNPs. We assumed that SNPs in these correlated signal regions are causal variants shared between the two populations and the effect correlation was set to be 1. The heritability was set to be 0.5 in both populations. We performed GWAS on 50,000 Europeans and 10,000 East Asians. We used the top 100 positive correlated regions as local genetic correlation annotation in X-Wing and repeated the simulation 20 times. For both European PRS and linearly combined PRS, X-Wing outperforms PRS-CSx and XPASS across all replicates (**Supplementary Figure 19**). These simulation results are consistent with the conclusion from our extensive real data analysis, demonstrating the statistical advance in X-Wing compared to existing approaches.

Supplementary Figure 19. Predictive accuracy (measured by R^2) of X-Wing, PRS-CSx, and XPASS in simulations. The boxes represent the percentages relative increase of R^2 over PRS-CSx across 20 simulation replicates. The PRS is (a) European PRS (b) Linearly combined PRS.

Minor comments

(1) Recent studies show that incorporating functional annotations represent different biological functionalities increase the association analysis power². This reviewer recommended the reviewer discuss the ability of X-Wing to incorporate multiple functional annotations of different biological aspects to further improve the accuracy, not only the epigenetics scores.

Reference

1. Li, X. et al. Dynamic incorporation of multiple in silico functional annotations empowers rare variant association analysis of large whole-genome sequencing studies at scale. *Nature genetics* 52, 969-983 (2020).

Response: Thanks for the suggestion. X-Wing can incorporate arbitrary sets of annotation data from different biological aspects, not only the epigenetic annotations. We have revised the manuscript as follows (page 13 in the revised manuscript).

“But we note that it is a general framework that can incorporate arbitrary sets of annotation data, such as the epigenetic annotations used in the PRS literature, *in silico* variant annotations based on machine learning exercises, or LD and allele frequencies which have been shown to improve heritability estimation³⁻⁸ (**Supplementary Note**).”

Reviewer #2 (Remarks to the Author):

In this paper, the Miao et al introduced the X-wing method, a novel PRS algorithm for cross-ancestry analyses. While X-wing only provide a marginal improvement over PRS-CSx, the authors introduce a procedure which allow parameter optimization without requiring additional genotype data. I do have a number of questions:

1. Judging from the equations, the main difference between X-wing and PRS-CSx is the addition of annotation-dependent shrinkage parameter, correct?

Response: X-Wing has three main advances compared to PRS-CSx. First, as the reviewer pointed out, we introduced an annotation-dependent statistical shrinkage approach which allows integrative analysis of functional annotation data in multi-population PRS modeling. This is a flexible framework that can incorporate arbitrary sets of annotation data to improve the PRS performance. Second, to maximize PRS portability between ancestral populations, we designed and implemented the cross-population local genetic correlation annotation for cross-population genetic risk prediction. This particular annotation is designed to identify correlated genetic effects between populations. When coupled with the annotation-dependent statistical shrinkage approach, the method amplifies SNP predictors that show correlated effects (and are thus portable) between populations. Third, we introduced a novel strategy to linearly combine the multi-ancestry PRS using the training GWAS summary statistics alone as input. In existing approaches such as PRS-CSx, additional individual-level genotype and phenotype data are required to fit a regression to combine multiple PRS. These data need to be from the non-EUR population and also need to be independent from the input GWAS as well as testing samples for PRS application. In practice, this type of data can be very difficult to obtain.

All three points listed above are major methodological advances that involve non-trivial statistical techniques. These advances have also been highlighted in Figure 1 in our manuscript.

2. One of my main concern is that the authors only simulated using chromosome 22, which means the simulated heritability will be disproportionately large compared to realistic scenarios. In addition, based on my experience, when only simulating 1 chromosome, especially if it is on chr22, the sample generated by HapGen2 will tends to be highly related. A more realistic simulation should be across all the whole genome.

Response: Thank you for the constructive suggestion. We have added new simulations based on genome-wide data in our revised manuscript. Genotypes for 50,000 individuals and 831,636 HapMap3 SNPs across the whole genome were simulated using the HapGen2. We simulated two independent traits for two populations under the infinitesimal model and assessed the type-I error rates for X-Wing and PESCA. X-Wing showed well-controlled type-I error rates across different heritability settings, but PESCA suffered substantial type-I error inflation (**Supplementary Table 5**). Further, we compared the statistical power under the heritability enrichment model. We randomly selected 50 genome segments, each spanning 1,000 SNPs, as the correlated signal regions. 30% trait heritability was

attributed to the signal regions and the other 70% was attributed to the rest of the genome. Correlation of SNP effect sizes in the correlated signal regions was set as 0.9. X-Wing achieved universally higher true positive rates than PESCA (**Supplementary Figure 2**). Of note, false positive rate for PESCA was larger than 0.4 across different trait heritability (**Supplementary Table 6**), suggesting that inferior statistical power is not contradictory with inflated type-I error for PESCA.

Supplementary Table 5. Type I error rates under the infinitesimal model for whole genome SNPs.

Trait heritability	X-Wing	PESCA
0.1	0.01	0.81
0.3	0.04	0.89
0.5	0.02	1

Supplementary Figure 2. True positive rates in simulations under heritability enrichment model for whole genome SNPs.

Supplementary Table 6. False positive rates under the heritability enrichment model for whole genome SNPs.

Trait heritability	X-Wing	PESCA
0.1	0.004	0.583
0.3	0.006	0.48
0.5	0.002	0.444

3. Just a note, when simulating using HapGen2, the F_{st} between the two subpopulation will tends to be much smaller than what was expected. E.g. simulated AFR vs EUR samples might only have a F_{st} of 0.006 instead of >1 , which might worth pointing out in the limitation

Response: Thank you for the suggestion. In the revised manuscript, we have added the limitation that HapGen2-simulated genotypes tend to have smaller F_{st} than expected between populations (page 14 in the revised manuscripts).

“Third, our simulations were carried out using HapGen2-simulated genotypes, which is known to have smaller fixation index (F_{ST}) than expected between two populations.”

4. In this paper, the authors mainly focuses on East Asian and European samples. However, when compared to African samples, the transferability of European based PRS to East Asian are usually easier. It would be worthwhile for the authors to also consider applying X-wing in the AFR samples similar to the PRS-CSx paper

Response: This is a great point. Following the reviewer’s suggestion, we have added new analyses to compare X-Wing’s PRS performance with other methods for 13 traits in 6,490 samples with African ancestry. We used UKB, BBJ, and PAGE GWAS as training data and European, East Asian, and admixed American LD reference panels. The decision to pair PAGE GWAS with admixed American LD reference is based on the fact that PAGE GWAS consists primarily of Hispanic/Latino. The linearly combined X-Wing PRS showed an average R^2 increase of 4.8% ($P_{\text{wilcoxon}} = 0.01$) compared to PRS-CSx in the African population (**Supplementary Figure 10c**). We also note that the X-Wing linear combination approach based on GWAS summary statistics showed nearly identical results compared to the linearly combined PRS using the gold-standard regression approach trained on individual-level data (regression slope=0.971) (**Supplementary Figure 10d**). If no external individual-level data are available for regression model training, X-Wing PRS showed a substantial improvement in prediction accuracy with the R^2 increase ranging from 40.5%-144.5% ($P_{\text{wilcoxon}} = 0.01$ to $2.4e-4$) in Africans (**Supplementary Figure 10e-g**). To conclude, these new results in the African populations are consistent with our previous findings obtained from East Asian and Admixed American populations. X-Wing outperforms PRS-CSx when applied to African ancestry samples.

Supplementary Figure 10. Performance of X-Wing for 13 traits in African ancestry samples. Panel a, b, and c show the percentage increase in R^2 of X-Wing over PRS-CSx for European, East Asian,

and linearly combined PRS. Panel **d** compares the R^2 for linearly combined PRS with mixing weights obtained using GWAS summary statistics and individual-level data. Panel **e**, **f**, and **g** show the percentage increase in R^2 of X-Wing PRS over PRS-CSx using only GWAS summary statistics. X-Wing+ PRS is the linearly combined X-Wing PRS using weights estimated using GWAS summary statistics. PRS-CSx PRS is calculated based on posterior mean effects of European, East Asian, and admixed American population, respectively. The dashed line represents the average increase.

5. Simulations were only done to compare the PESCA and X-wing, given the main purpose of X-wing, a more interesting simulation might be to compare the performance between X-wing and PRS-CSx to investigate in what scenarios PRS-CSx performs outperform X-wing and vis versa.

Response: Thanks for the suggestion. We have added new simulations to compare the predictive accuracy (measured by R^2) of X-Wing with two existing methods, PRS-CSx and XPASS. We used HapGen2-simulated genotypes for Europeans and East Asians released in Zhang et al.¹, which contains 956,041 Hapmap3 SNPs after quality control. We randomly selected 50 genomic segments, each spanning 1,000 SNPs. We assumed that SNPs in these correlated signal regions are causal variants shared between the two populations and the effect correlation was set to be 1. The heritability was set to be 0.5 in both populations. We performed GWAS on 50,000 Europeans and 10,000 East Asians. We used the top 100 positive correlated regions as local genetic correlation annotation in X-Wing and repeated the simulation 20 times. For both European PRS and linearly combined PRS, X-Wing outperforms PRS-CSx and XPASS across all replicates (**Supplementary Figure 19**). These simulation results are consistent with the conclusion from our extensive real data analysis, demonstrating the statistical advance in X-Wing compared to existing approaches.

Supplementary Figure 19. Predictive accuracy (measured by R^2) of X-Wing, PRS-CSx, and XPASS in simulations. The boxes represent the percentages relative increase of R^2 over PRS-CSx across 20 simulation replicates. The PRS is (a) European PRS (b) Linearly combined PRS.

6. For the cross validation procedure, my understanding is that the author split the target samples (which can be from, say East Asian) in half, use half of those sample for GWAS, and then use the calculated summary statistics and the summary statistics

from a reference cohort (e.g. from European) to calculate a PRS in the 1000 Genome (with matched population, eg. calculate EUR PRS in EUR 1000G, calculate X PRS in X 1000G where X is the target population). Finally, the weights were estimated using the genotype and phenotypes of the remaining target samples (the validation cohort), correct? In that case, what is the sample size limitation for this method to work? I can imagine that if the target cohort size is small e.g. < 1000, which is usually the case for under-representative populations, then the CV procedure might be highly unstable?

Response: Thanks for the question. To clarify, we did not use any samples from the target cohort to do cross-validation. The target cohort is only used for benchmarking the R² for different PRS methods. Instead, our approach can be viewed as doing repeated learning in the input GWAS samples, which typically have a huge sample size to ensure the performance of the cross-validation.

For example, when using GWAS summary statistics from UKBB and BBJ as input, our approach can be viewed as doing repeated learning in BBJ samples (N = 42,790 ~ 159,095). For 4-fold repeated learning, our method can be viewed as the following steps with the individual-level genotype and phenotype:

- 1) Split the BBJ samples into 75% training and 25% validation sets.
- 2) Run GWAS on 75% training sets samples to obtain the training GWAS summary statistics.
- 3) Use this 75% training BBJ GWAS summary statistics and full UKBB GWAS summary statistics as input for X-Wing and obtain the population-specific SNP posterior effects.
- 4) Generate population-specific PRS in the 25% validation sets and estimate the linear combination weights.
- 5) Repeat the above (1)-(4) four times and calculate the averaged linear combination weights.
- 6) Apply this averaged linear combination weights to linearly combine the population-specific PRS using full UKBB and BBJ GWAS summary statistics as input for X-Wing.

Notably, we showed that we only require the summary statistics from the input GWAS sample (like BBJ) and LD reference panel to complete this cross-validation. We do not need any individual-level genotype and phenotype data throughout the process.

To conclude, our approach mimics the cross-validation in the input GWAS sample, which typically has a large sample size to guarantee the performance of the CV procedure. We have improved the writing in the manuscript to make the CV procedure clearer (page 18-19 in the revised manuscripts).

“Generalizing the technique in our recent work⁹ and its extension handle the LD⁴, we introduce a summary statistics-based repeated learning strategy, which mimics the individual-level repeated learning but does not need individual-level GWAS data (**Supplementary Note**). This approach has three main steps which we describe below. Since this approach does not involve a separate validation sample, we will perform analysis using input GWAS from the target population (e.g. BBJ GWAS when East Asian is the target population), the sample size of which is typically sufficiently large

to ensure the performance of repeated learning. Without loss of generality, we denote $k = 1$ for this (target) population.”

7. Follow up on that, in the paper, the authors only perform 4 fold cross validation, which seems like a relatively small number. Would that be enough? How does the number of fold affects the stability of the algorithm? I'd imagine that will be a function of the sample size (e.g. number of cases) and the number of folds?

Response: Thanks for the comment. In our primary analysis, the linearly combined PRS based on weights obtained from 4-fold summary statistics-based repeated learning already showed nearly identical performance compared to PRS constructed from individual-level data, which we considered the gold standard (**Figure 5b** and **Supplementary Figure 20a**). These suggest that 4-fold repeated learning should be sufficient to combine multi-ancestry PRS. To investigate whether adding more folds would further improve the prediction accuracy, we have added new analyses in the revision to benchmark the prediction accuracy of lineally combined PRS using weights obtained from 10-fold repeated learning for 31 traits in East Asians. The predictive R^2 of X-Wing linearly combined PRS using weights obtained from 4-fold and 10-fold repeated learning is very similar and statically not distinguishable (regression slope = 1.01, $P_{wilcoxon} = 0.14$) (**Supplementary Figure 31**). Nevertheless, in our software implementation, we allow the users to specify the number of folds in repeated learning in case it is a concern in some applications.

Supplementary Figure 31. Impact of the number of repeated learning folds on prediction accuracy of X-Wing linearly combined PRS with mixing weights obtained using GWAS summary statistics for 31 traits in East Asians. X-axis represents 4 folds and Y-axis is 10 folds repeated learning. The P-value of two-sided Wilcoxon signed-rank test is $P_{wilcoxon} = 0.14$.

8. My other concern regarding this procedure is that the reference population (e.g. the 1000 Genome) tends to ascertain for healthy samples. Would that lead to bias to your algorithm? How similar will the weight selected from this CV procedure compared to if we have an independent sample for optimization?

Response: Thanks for the question. To clarify, the only reason we need reference panels (e.g., the 1000 Genomes) is to provide the LD for the input GWAS sample. This is a common strategy for training PRS models without in-sample LD. Thus, we

do not expect this would lead to bias in the algorithm. Moreover, we have shown that the prediction accuracy of our linear combined PRS is almost the same compared to the gold-standard regression approach using an independent sample (**Figure 5b** and **Supplementary Figure 20a**). This further confirms that our approach is robust to the use of external LD.

But just to be certain, we followed the reviewer’s suggestion and compared the weights selected from our repeated learning procedure and the weights estimated from an independent sample. The weights selected from our repeated learning procedure for 29/31 traits fall into the 95% confidence interval of the weights estimated in an independent sample (**Supplementary Figure 30**).

Supplementary Figure 30. Comparison of linear combination weights estimated from summary statistics and individual-level validation data for 31 traits in East Asians. The black point represents the mean of estimated weights, and the error bar represents the 95% confidence interval across 100 replicates in an independent sample. The pink cross point is the weights estimated from summary statistics-repeated learning.

We have revised the relevant text to clearly state that our method only uses LD information in the reference panel (page 19 in the revised manuscript).

“We note that when we calculate $\hat{\mathbf{w}}$ using PRS matrix in the reference panel, essentially only LD matrix is used: $\mathbf{PRS}^{(v)T} \mathbf{PRS}^{(v)} = \mathbf{b}^T \mathbf{X}^{(v)T} \mathbf{X}^{(v)} \mathbf{b} \approx \frac{N_1^{(v)}}{N^{(ref)}} \times \mathbf{b}^T \mathbf{X}^{(ref)T} \mathbf{X}^{(ref)} \mathbf{b} = \frac{N_1^{(v)}}{N^{(ref)}} \times \mathbf{PRS}^{(ref)T} \mathbf{PRS}^{(ref)}$, where $\frac{\mathbf{X}^{(ref)T} \mathbf{X}^{(ref)}}{N^{(ref)}}$ is the LD matrix. We choose to calculate $\hat{\mathbf{w}}$ using PRS matrix to reduce computational complexity compared to directly using LD matrix, but one can still estimate $\hat{\mathbf{w}}$ using only LD matrix in the reference panel (**Supplementary Note**).”

9. Why does the authors exclude the negatively correlated regions? Was there an implicit assumption that the direction of effect should be consistent across populations? Would relaxing this assumption negatively impact the performance of X-wing?

Response: Thanks for the question. We aimed to amplify SNP effects that are more portable between the source population and the target population. Thus, we felt it would not make sense to amplify the effects in negatively correlated regions. To verify our claims, we have added the PRS analysis by using both positively and negatively correlated regions as annotations for 31 traits in East Asians. X-Wing with positive and negative regions achieves worse performance compared to X-Wing with only positive regions (**Panel a and b**), but still outperforms PRS-CSx (**Panel c and d**). This suggests that excluding the negatively correlated regions as portable annotations improves the prediction accuracy of PRS. Regardless of how these negatively correlated regions are handled, the main conclusions in our manuscript remain unchanged.

Supplementary Figure 27. Impact of using both positively and negatively correlated regions as annotation on PRS prediction accuracy. (a) X-Wing + both positively and negatively correlated region vs X-Wing + only positively correlated region European PRS ($P_{wilcoxon} = 2.8e - 5$). (b) X-Wing + both positively and negatively correlated region vs X-Wing + only positively correlated region linearly combined PRS ($P_{wilcoxon} = 0.02$). (c) X-Wing + both positively and negatively correlated vs PRS-CSx European PRS ($P_{wilcoxon} = 5.4e - 4$). (d) X-Wing + both positively and negatively correlated vs PRS-CSx linearly combined PRS ($P_{wilcoxon} = 9.2e - 6$). The dashed line represents the average decrease.

10. Does the chose of upper bound of region size, and/or the merge distance, and/or the s parameter affects X-wing's performance?

Response: Thanks for the suggestive comment. We have performed additional sensitivity analyses of GWAS data from UKB and BBJ. X-Wing achieved highly concordant results with different upper bound of region size (**Supplementary Figure 14-15**) and different merge distance (**Supplementary Figure 16**). We did not perform the PRS analysis with different merge distance because the local-genetic correlation annotation derived remains unchanged.

Supplementary Figure 14. Number of significant regions identified by X-Wing for 31 complex traits with different upper bound of region size.

Supplementary Figure 15. Comparison of the prediction accuracy between X-Wing and PRS-CSx PRS for 31 traits in East Asians with varying upper bound of region size. Panel a-d: X-Wing PRS

with upper bound of region size=1000. Panel e-h: X-Wing PRS with upper bound of region size=3000. The baseline in panel e-h is the PRS-CSx. The P-value of two-sided Wilcoxon signed-rank test for analysis in each panel are (a)0.92; (b)0.90; (c)0.45; (d)0.26; (e)2.6e-6; (f)5.0e-7; (g)4.0e-6; (h)5.0e-7.

Supplementary Figure 16. Number of significant regions identified by X-Wing for 31 complex traits with different merge distance.

Minor comments:

1. When stating "gain in predictive R2" should specify that these are "relative gain"

Response: Thanks for the comment. We have revised it accordingly.

2. for region size, is that the number of variants or size of a region (e.g. 2000 variants or 2000 bp?)

Response: Region size means the number of SNPs in the region. We have clarified it in the revised manuscript.

3. Given the addition complexity, and considering that PRS-CSx isn't known for its speed, how computationally intensive is X-wing?

Response: Thanks for the suggestions. We have added descriptions of the computation cost and requirements for X-Wing (page 18 in the **Supplementary Note**).

"X-Wing by default runs analysis on each chromosome and estimates the posterior mean effects for each population in parallel. The computational demands of X-Wing depend on the number of SNPs in GWAS summary statistics. For our analysis using UKBB and BBJ summary statistics as discovery GWAS and UKBB East Asian samples as test data, it takes about 1.5 hours to finish estimating posterior effects in chromosome 1 and about 10 minutes in chromosome 22 with 2000 MCMC iterations (default), using single thread in Intel Xeon Gold E5-4620 processor (2.60 GHz).

We further compared the computation cost and requirements of X-Wing, PRS-CSx, and XPASS using the same machine described above. XPASS does not allow parallel

computation but only takes ~10 minutes to finish the computing for all 22 chromosomes. Both PRS-CSx and X-Wing allow parallel computing over chromosomes. For the largest chromosome (chromosome1), PRS-CSx takes about 1 hour to finish the computation and has similar memory usage as X-Wing (**Supplementary Table 20**). Since X-Wing requires estimating the posterior effects for each population separately, the number of parallel computing jobs of X-Wing is K times of PRS-CSx, where K is the number of GWAS summary statistics included.”

Reviewer #3 (Remarks to the Author):

Miao and colleagues have developed a method called X-Wing with the aim to improve PRS transferability across ancestries. This is an important research question to achieve equitable benefits and uses of PRS in global populations. X-Wing uses information from between-ancestry local genetic correlation as functional annotations to estimate population-specific posterior mean effects. The authors ran both simulations and real data analyses to explore the reliability of identifying genomic regions with significant local genetic correlation. UKB, BBJ and PAGE datasets on various complex traits were further used to construct PRS using X-Wing as compared to other methods (PRS-CSx, PolyFun-Pred, XPASS). Regarding the method, a lot of extensions have been done on integrating the authors' previous work. The manuscript is overall well organized (with a well-documented github page). I have some comments below that hopefully improve the manuscript.

Major comments:

1) The strategy to detect regions showing local genetic correlation is an extension to the authors' previous work LOGODetect. But it is unclear how the method can be applied for more than two ancestral populations when the authors using UKB, BBJ and PAGE as the training datasets. Also, would high LD regions (such as MHC region) affect this and following analyses?

Response: Thank you for the comment. We ran LOGODetect for population pairs separately when applying for more than two ancestral populations. For example, when using UKB, BBJ, and PAGE as the training datasets and admixed Americans as the testing sample, we performed LOGODetect between UKB and PAGE, as well as BBJ and PAGE, to obtain the local genetic correlation annotation between Europeans and admixed Americans, and between East Asians and admixed Americans, respectively.

We have also added the analysis with the MHC region excluded from local genetic correlation estimation. The prediction accuracy of PRS is very similar to the analysis that included the MHC region (**Supplementary Figure 17**).

Supplementary Figure 17. Impact of excluding MHC region in identifying regions with local genetic correlation on PRS prediction accuracy for 31 traits in East Asians. (a) X-Wing (annotation that excludes MHC region) vs X-Wing (annotation that includes MHC region) European PRS: ($P_{wilcoxon} = 0.22$). (b) X-Wing (annotation that excludes MHC region)+ vs X-Wing (annotation that includes MHC region) linearly combined PRS: ($P_{wilcoxon} = 0.09$). (c) X-Wing (annotation that excludes MHC region) vs PRS-CSx European PRS ($P_{wilcoxon} = 7.0e - 05$). (d) X-Wing (annotation that excludes MHC region) vs PRS-CSx linearly combined PRS ($P_{wilcoxon} = 1.2e - 05$). All annotations are based on top 1000 positive regions.

We have revised the manuscript as follows (page 17 in the revised manuscript).

“Without loss of generality, we assume population 1 is the target population. We break down our algorithm into three steps:

Step1: Obtain annotation information through local genetic correlation analysis

We perform local genetic correlation analysis between population 1 and population k ($k = 2, \dots, K$) to identify top s regions with positive local genetic correlation. We denote the set of regions as Ω_k (e.g., when using UKB, BBJ, and PAGE as training GWAS, we ran local genetic correlation analysis between UKB and PAGE, as well as between BBJ and PAGE). We selected $s = 1000$ in our primary analysis and demonstrated that PRS performance is robust to the choice of s (**Supplementary Figure 12 and 13**). We also used regions with both positive and negative local genetic correlation as annotation and demonstrated that the PRS performs better when only positive regions are used (**Supplementary Figure 27**).”

2) For the annotation-dependent Bayesian horseshoe regression model:

a. Categorical annotations were used in the study. As the region identified, it would be possible and reasonable to calculate the values of local genetic correlation. I was wondering whether using the specific values would further contribute to the PRS performance.

Response: Thank you for the comment. Currently, there is no method that can directly estimate local genetic correlation in a small region between two populations. The scan statistic defined in our paper can reveal whether two populations are correlated in a given region, with a large statistic value deviating from the null distribution representing significant local genetic correlation. However, the null distributions of scan statistics are different across genome regions, which makes scan statistic values not comparable between each other. Therefore, the scan statistic we used in this paper is not an appropriate estimator for local genetic correlation and therefore we hesitate to use their specific values for PRS calculation. There also exist some methods (e.g. ρ -HESS¹⁰ and SUPERGENOVA¹¹) in the literature that can estimate local genetic correlation between two traits in the same population. While it may be possible to extend these approaches to estimate cross-ancestry local genetic correlation, the estimates produced by these approaches tend to be noisy in many regions in the genome. We consider estimation for local genetic correlation (between two populations or between two traits) to remain a challenging problem, and we leave incorporating specific value of local genetic correlation into PRS construction a future research direction.

b. Marginal effects were assumed in the model. As most current GWAS (such as BBJ used in this study) are performed with mixed models, explorations or comments on how this will affect the model performance would be helpful.

Response: We appreciate the comment. Mathematically, the GWAS association results using the linear mixed model is equivalent to the results using the marginal linear model on phenotypic residual after adjusting for best linear unbiased prediction (BLUP)^{12,13}. This phenotypic residual can be considered the phenotype after adjusting for the sample relatedness (genetic relationship matrix). In our analysis, the BBJ GWAS are performed using BOLT-LMM. BOLT-LMM utilized a two-step approach to conduct the linear mixed model GWAS: “Our algorithm fits a Gaussian mixture model of SNP using a fast variational approximation to compute approximate phenotypic residuals and tests the residuals for association with candidate markers via a retrospective score statistic”. Therefore, the mixed model GWAS from BOLT-LMM is from the marginal linear regression between the phenotypic residuals and the SNP. To summarize, most current GWAS (such as BBJ used in this study) essentially comes from a marginal linear model with phenotypic residuals as outcome and SNP as the predictor. Therefore, we believe our marginal effects assumption is valid, and we do not expect it would influence the model performance. We have clarified this point in the revised manuscript (page 16 in the **Supplementary Note**).

“X-Wing assumes a marginal linear regression between the phenotype and SNP. In practice, many of the GWAS are performed using the mixed model. Mathematically, the GWAS association results using the linear mixed model is equivalent to the results using the marginal linear model on phenotypic residual after adjusting for best linear unbiased prediction (BLUP)^{12,13}. This phenotypic residual can be considered the phenotype after adjusting for the sample relatedness (genetic relationship matrix). In our analysis, the BBJ GWAS are performed using BOLT-LMM. BOLT-LMM utilized a two-step approach to conduct the linear mixed model GWAS: “Our algorithm fits a Gaussian mixture model of SNP using a fast variational approximation to compute approximate phenotypic residuals and tests the residuals for association with candidate markers via a retrospective score statistic”. Therefore, the mixed model GWAS from BOLT-LMM is from the marginal linear regression between the phenotypic residuals and the SNP. To summarize, most current GWAS (such as BBJ used in this study) essentially comes from a marginal linear model with phenotypic residuals as outcome and SNP as predictor. Therefore, our marginal effects assumption is valid, and we don’t expect it will influence the PRS model performance. And our repeated learning approach can be considered as splitting on this phenotypic residual instead of the raw phenotype that is correlated among correlated individuals. Therefore, applying the summary statistics-based splitting using summary statistics that have already been accounted for sample relatedness should not cause problems.”

c. I was not convinced by the rationale that annotation-dependent shrinkage was not applied for GWAS with same ancestry as target populations because there is no loss of portability. Would implementing annotations further improve the PRS performance for target population when using target-ancestry specific effects?

Response: Thanks for the question. In principle, one can employ annotation-dependent shrinkage for target population based on arbitrary annotation data. Examples of annotation include various epigenetic marks, *in silico* variant annotations based on machine learning exercises, or LD and allele frequency information which have been shown to improve heritability estimation. It is true that some annotation may help improve PRS performance in the target population. In fact, we and others have published on this topic^{3,14,15}. Therefore, we understand the reviewer's point about there could be benefits to also applying annotation-dependent shrinkage for the target population.

However, we note that the annotation we introduced in the manuscript is cross-population local genetic correlation. The rationale of using this annotation is to directly identify genomic regions with correlated effects between populations, and then amplify these correlated and portable effects in PRS models. Using a joint analysis of European and East Asian GWAS as an example (East Asian is the target population), here the local correlation annotation will help stratify genomic regions into 1) regions with correlated effects between Europeans and East Asians and 2) regions with population-specific associations. When training a PRS using European GWAS, we want to amplify SNP effects that are more portable (i.e., located in the annotation regions) and downweigh the SNPs that are European-specific. But this strategy will not make sense when training a PRS using East Asian GWAS since Asian-specific effects and correlated effects between Asians and Europeans are equally important for predicting the phenotype in East Asians. This is the reason why we do not employ any annotation-dependent shrinkage when estimating posterior SNP effects for the target population.

Finally, we note that we allow the users to incorporate any annotations they prefer in our implemented software. They can modify these choices about shrinkage depending on the annotations used in the analysis.

3) For linearly combine multiple PRS:

a. Related to previous marginal effects assumption, most GWAS are performed accounting for sample relatedness. Therefore, I think it will not be reasonable to apply splitting for such GWAS, although the results show the accuracy was not affected. The authors should account for this or justify its use in such scenarios.

Response: We appreciate the comment. As mentioned in the response to the previous question of marginal effects assumption, the GWAS association results using linear mixed model is equivalent to the results using the marginal linear model on phenotypic residual after adjusting for sample relatedness.^{12,13} Thus, our splitting approach on LMM-based summary statistics can be considered as splitting on this phenotypic residual instead of the raw phenotype. In addition, as mentioned by the reviewer, the empirical prediction accuracy was not affected by the fact that input GWAS associations were obtained from LMM, which further validates our approach with LMM-based summary statistics. We have clarified this point in the revised manuscript (page 16 in the **Supplementary Note**).

“X-Wing assumes a marginal linear regression between the phenotype and SNP. In practice, many of the GWAS are performed using mixed model. Mathematically, the GWAS association results using linear mixed model is equivalent to the results using marginal linear model on phenotypic residual after adjusting for Best linear unbiased prediction (BLUP)^{12,13}. This phenotypic residual can be considered the phenotype after adjusting for the sample relatedness (genetic relationship matrix). In our analysis, the BBJ GWAS are performed using BOLT-LMM. BOLT-LMM utilized a two-step approach to conduct the linear mixed model GWAS: “Our algorithm fits a Gaussian mixture model of SNP using a fast variational approximation to compute approximate phenotypic residuals and tests the residuals for association with candidate markers via a retrospective score statistic”. Therefore, the mixed model GWAS from BOLT-LMM is from the marginal linear regression between the phenotypic residuals and the SNP. To summarize, most current GWAS (such as BBJ used in this study) essentially comes from a marginal linear model with phenotypic residuals as outcome and SNP as predictor. Therefore, our marginal effects assumption is valid, and we don’t expect it will influence the PRS model performance. And our repeated learning approach can be considered as splitting on this phenotypic residual instead of the raw phenotype that is correlated among correlated individuals. Therefore, applying the summary statistics-based splitting using summary statistics that have already been accounted for sample relatedness should not cause problem.”

b. I am confused about how the use of LD reference panel would result in overfitting. The authors divide the small 1000G cohorts to even smaller datasets, which overall could reduce the precision of LD approximation. Also, the authors mentioned in the discussion that the mismatch between LD reference panel and discovery GWAS could impact the PRS performance, especially for multi-ancestry GWAS such as PAGE. Descriptions about ancestry proportions of PAGE would be helpful, also please add what LD reference was used for PAGE. However, the impact of such LD mismatch was not explored here as 1000G is also an external LD reference panel. As X-Wing is dependent on LD reference panels, I think further exploration about the impact of them would be important to show the results won’t be largely affected by them.

Response: We thank the reviewer for the comment. Ideally, the dataset used for training PRS and the dataset used for combining multiple scores should be independent of each other. Otherwise, PRS in the second dataset could be overfitting which may lead to unreliable results for regression weights. In X-Wing, we used the LD reference panel to approximate the in-sample LD. To make sure the independence of training and holdout datasets, we decided to divide the 1000G reference samples into subsets and coupled them with separate GWAS datasets in different stages of the analysis. Moreover, we have shown that the prediction accuracy of our linear combined PRS based on this strategy is almost identical compared to the gold-standard regression approach using an independent sample. This indicates that division of the 1000G reference did not negatively impact our model performance.

In PAGE studies (n= 49,839), the genotyped self-identified as Hispanic/Latino (n = 22,216), African American (n = 17,299), Asian (n = 4,680), Native Hawaiian (n = 3,940), Native American (n = 652) or Other (n = 1,052). Since the PAGE study comprised

largely African American and Hispanic/Latino samples, we followed the PRS-CSx paper and coupled PAGE GWAS with the 1KG AMR reference panel in the PRS analyses. To investigate the impact of using different LD reference panels on prediction accuracy, we have added an analysis in which we changed the 1KG AMR into 1KG AFR or EUR reference panel when doing the PRS analysis for 13 traits in Africans. As expected, using either 1KG AFR or EUR reference leads to reduced prediction accuracy compared with using AMR reference panel (**Supplementary Figure 29**). This indicates that using 1KG AMR reference shows the best performance and it is important to choose the LD reference panel that best matches the population of the GWAS samples in practice.

Supplementary Figure 29. Impact of mismatched LD reference panel on prediction accuracy for 13 traits in Africans. The baseline method is the X-Wing PRS using admixed American (AMR) LD reference panel. The methods in each figure are **(a)** X-Wing admixed American PRS using EUR LD reference panel (The P-value of two-sided Wilcoxon signed-rank test is $P_{wilcoxon} = 2.4e - 4$). **(b)** X-Wing linearly combined PRS using EUR LD reference panel ($P_{wilcoxon} = 2.4e - 4$). **(c)** X-Wing admixed American PRS using AFR LD reference panel ($P_{wilcoxon} = 0.017$). **(d)** X-Wing linearly combined PRS using AFR LD reference panel ($P_{wilcoxon} = 1.2e - 3$). The dashed line represents the average decrease.

We have added the descriptions about ancestry proportions of PAGE (page 22 in the revised manuscript) as well as the detailed number of samples from each population in the **Supplementary Table 7**.

4) Regarding the accuracy comparisons among different methods, they were not intuitive and informative, specifically:

a. The accuracy is mainly reported based on relative improvement without standard errors, CIs or p-value estimates, thus it is hard to conclude whether the improvement is significant here. For example, the 4.1% increase in R² compared to PRS-CSx seems to be only a marginal increase based on the number. Please add such statistics.

Response: This is a great point. We have added P-values of the two-sided Wilcoxon signed-rank test in the revised manuscript to compare the predictive accuracy of X-Wing with other methods in all analyses. All tests were statistically significant ($P < 0.05$), indicating that X-Wing shows significant improvement over PRS-CSx, XPASS, PolyFun-Pred, and PolyFun+ in non-European populations.

We have included p-values in all places where the percentage increase in R^2 is reported.

b. The authors report the accuracy using EUR-based PRS and linearly combined multi-ancestry PRS using X-Wing in two sections, respectively. It would be helpful to see the direct comparisons between the two PRS. If the linear combination of PRS did not significantly improve the accuracy, then it might suggest we can directly use the population specific PRS without combine them; if it increased, then comments on this would be helpful.

Response: We appreciate the comment. We have added **Supplementary Figure 23** to directly compare the prediction accuracy between EUR-based PRS and linearly combined multi-ancestry PRS using X-Wing. For all approaches, linear combination (LC) significantly improves the PRS accuracy (by comparing EUR and LC in the X-axis in **Supplementary Figure 23**). The improvement in prediction accuracy is expected because it uses additional data to train the optimal linear combination of multi-ancestry PRS by minimizing the mean squared error between the linearly combined PRS and phenotype from target population.

Supplementary Figure 23. Benchmark different methods' performance in different test sample.

The test sample are **(a)** 31 traits in East Asians. **(b)** 16 traits in East Asians with PolyFun-Pred PRS coefficient available. **(c)** 13 traits in admixed Americans. **(d)** 13 traits in Africans. UKB and BBJ GWAS summary statistics are used as training data when the test sample are East Asians; UKB, BBJ and PAGE summary statistics are used as training data when the test sample are admixed Americans or Africans. “(EUR)” represents the PRS based on European posterior mean effects and “(LC)” represents the linearly combined PRS. “X-Wing (LC)” and “X-Wing+ (LC)” represent the X-Wing linearly combined PRS using mixing weights estimated from individual-level data and summary statistics-based repeated learning, respectively. “PolyPred+ (LC)” linearly combines the effect sizes of BOLT-LMM-UKB, PolyFun-Pred, and BOLT-LMM-BBJ.

c. Directly comparing the accuracy of different methods in the same figure would be more intuitive and informative. Specifically, for PolyFun-Pred, the extensions of summary-level based PolyPred by integrating multiple populations/predictors and a few other methods with available coefficients would be worth to be compared.

Response: We thank the reviewer for the suggestion. We have added the figure directly comparing the accuracy of different methods. We also added the PolyPred+ (which linearly combines PRS from PolyFun-pred, BOLT-LMM-UKB, and BOLT-LMM-BBJ) in our analysis for 31 traits in East Asians. Overall, X-Wing PRS shows better predictive performance over alternative methods tested for all analyses (**Supplementary Figure 23**).

Supplementary Figure 23. Benchmark different methods' performance in different test sample. The test sample are **(a)** 31 traits in East Asians. **(b)** 16 traits in East Asians with PolyFun-Pred PRS coefficient available. **(c)** 13 traits in admixed Americans. **(d)** 13 traits in Africans. UKB and BBJ GWAS summary statistics are used as training data when the test sample are East Asians; UKB, BBJ and PAGE summary statistics are used as training data when the test sample are admixed Americans or Africans. “(EUR)” represents the PRS based on European posterior mean effects and “(LC)” represents the linearly combined PRS. “X-Wing (LC)” and “X-Wing+ (LC)” represent the X-Wing linearly combined

PRS using mixing weights estimated from individual-level data and summary statistics-based repeated learning, respectively. “PolyPred+ (LC)” linearly combines the effect sizes of BOLT-LMM-UKB, PolyFun-Pred, and BOLT-LMM-BBJ.

d. The results show the improvement of X-Wing is largely dependent on trait-specific genetic architecture but without specific comments about this (except for BAS). Further, the traits here are all quantitative traits, thus it would be helpful to show a few disease traits.

Response: Following the reviewer’s suggestion, we have added type-2 diabetes as a disease trait to benchmark different methods. We used the European GWAS in Scott et al.¹⁶ and East Asian GWAS in Sunzuki et al.¹⁷ as training data. Both X-Wing European and linearly combined PRS shows better predictive performance than other methods in terms of liability R² and area under the ROC curve (AUC) (**Supplementary Figure 22**). X-Wing linear combined PRS with mixing weights obtained using GWAS summary statistics (X-Wing+ in the figure) shows similar performance with linear combined PRS using weights estimated from individual-level data and better performance than EUR-based PRS-CSx PRS.

Supplementary Figure 22. Comparison of the prediction accuracy for type-2 diabetes in East Asians between (a) Liability R² of European PRS (b) Liability R² of linearly combined PRS (c) Area under the ROC Curve (AUC) of European PRS (d) AUC of linearly combined PRS.

e. Not clear about the SNPs included: Only HapMap3 SNPs used? Or genome-wide imputed variants? How can SNP density impact the results (especially the local genetic correlation)?

Response: We used genome-wide imputed SNPs in the analysis of identifying regions with local genetic correlation and HapMap3 SNPs in the analysis of fitting the PRS model (following PRS-CSx to reduce memory and computational cost). We also performed additional sensitivity analyses of GWAS data from UKB and BBJ. When

using only HapMap3 SNPs, X-Wing identified consistent but slightly fewer significant regions than using genome-wide imputed SNPs (**Supplementary Figure 5**). In the second step of constructing PRS with local genetic correlation annotation, X-Wing uses only HapMap3 SNPs to reduce computation burden. We further performed additional PRS analysis using X-Wing with the annotation derived from regions using HapMap3 SNPs for 31 traits in East Asians. We compared its performance with X-Wing with annotation derived from regions using genome-wide imputed SNPs and PRS-CSx. X-Wing achieves consistent PRS results and outperforms PRS-CSx when using HapMap3 SNPs to construct annotation (**Supplementary Figure 5**). These suggest that X-Wing's superior performance is robust to the choice of SNPs included to identify regions with local genetic correlation.

Supplementary Figure 5. Number of significant regions identified by X-Wing using genome-wide imputed SNPs and HapMap3 SNPs for 31 complex traits.

Supplementary Figure 28. Impact of using hapmap3 SNPs in identifying regions with local genetic correlation on PRS prediction accuracy for 31 traits in East Asians. (a) X-Wing (Hapmap3 SNPs annotation) vs X-Wing (Genome-wide imputed SNPs annotation) European PRS ($P_{wilcoxon} = 0.08$). (b) X-Wing (Hapmap3 SNPs annotation)+ vs X-Wing (Genome-wide imputed SNPs annotation) linearly combined PRS ($P_{wilcoxon} = 0.06$). (c) X-Wing (Hapmap3 SNPs annotation) vs PRS-CSx European PRS ($P_{wilcoxon} = 1.7e - 5$). (d) X-Wing (Hapmap3 SNPs annotation) vs PRS-CSx linearly combined PRS ($P_{wilcoxon} = 3.5e - 6$). All annotations are based on top 1000 positive regions. The dashed line represents the average decrease.

5) Extensions of analyses to African populations would be very helpful as PRS has the lowest portability in them, especially when PAGE was added in the discovery GWAS. But here AMR with small sample sizes from UKB was focused.

Response: This is a great point. Following the reviewer's suggestion, we have added new analyses to compare X-Wing's PRS performance with other methods for 13 traits in 6,490 samples with African ancestry. We used UKB, BBJ, and PAGE GWAS as training data and European, East Asian, and admixed American LD reference panels. The decision to pair PAGE GWAS with admixed American LD reference is based on the fact that PAGE GWAS consists primarily of Hispanic/Latino. The linearly combined X-Wing PRS showed an average R^2 increase of 4.8% ($P_{wilcoxon} = 0.01$) compared to PRS-CSx in the African population (**Supplementary Figure 10c**). We also note that the X-Wing linear combination approach based on GWAS summary statistics showed nearly identical results compared to the linearly combined PRS using the gold-standard regression approach trained on individual-level data (regression slope=0.971) (**Supplementary Figure 10d**). If no external individual-level data are available for regression model training, X-Wing PRS showed a substantial improvement in prediction accuracy with the R^2 increase ranging from 40.5%-144.5% ($P_{wilcoxon} = 0.01$ to $2.4e-4$) in Africans (**Supplementary Figure 10e-g**). To conclude, these new results in the African populations are consistent with our previous findings obtained from East Asian and Admixed American populations. X-Wing outperforms PRS-CSx when applied to African ancestry samples.

Supplementary Figure 10. Performance of X-Wing for 13 traits in African ancestry samples. Panel a, b, and c show the percentage increase in R^2 of X-Wing over PRS-CSx for European, East Asian, and linearly combined PRS. Panel d compares the R^2 for linearly combined PRS with mixing weights obtained using GWAS summary statistics and individual-level data. Panel e, f, and g show the percentage increase in R^2 of X-Wing PRS over PRS-CSx using only GWAS summary statistics. X-Wing+ PRS is the linearly combined X-Wing PRS using weights estimated using GWAS summary statistics. PRS-CSx PRS is calculated based on posterior mean effects of European, East Asian, and admixed American population, respectively. The dashed line represents the average increase.

Minor comments:

1) In the introduction, “the third approach” or the third type of approaches?

Response: It is the third type of approaches. We have revised it accordingly.

2) Figure 2: please add descriptions for power and genetic covariance.

Response: Power is defined as the proportion of simulation repeats that the true signal region is identified. Genetic covariance measures the covariance of additive genetic component between two populations. We have added the description in the revised manuscript accordingly.

3) Figure 3C: please comment on why combining discovery and replication stage results in less identified regions? How could the number of regions affect by various factors?

Response: We applied X-Wing to study genetic correlation for four lipids traits independently in two datasets, using summary statistics in UKBB and BBJ as discovery dataset and those in GLGC and AGEN as replication dataset. We did not combine the two datasets together to perform analysis. To demonstrate the robustness of X-Wing across different datasets, we checked the extent of overlap between regions identified in two stages. Since sample sizes in GLGC and AGEN are

much smaller, regions identified in the replication stage were fewer than but shared substantial overlap with regions identified in the discovery stage.

4) Figure 5, why PRS-CSx is only based on EUR-based PRS when comparing to linearly combined PRS using X-Wing?

Response: One major innovation of our paper is that we introduce a method to linearly combine multiple PRS trained in various populations using GWAS summary data alone as input. PRS-CSx requires the individual-level validation data from the target population to estimate the linear combination weights and then linearly combine multi-ancestry PRS. If no external individual-level data are available for regression model training, the current best PRS approach in practice is to use posterior SNP effects estimated for one population. In **Figure 5**, we benchmarked our method with PRS-CSx under realistic scenarios where no individual-level validation set from target population is available. Therefore, without individual-level validation set, PRS-CSx can only produce population-specific PRS, while X-Wing can construct linearly combined PRS.

5) The results show that most genomic regions have positive genetic correlation, which are interesting and convincing. They are consistent with some previous studies that common causal variants are largely shared between-ancestries. The authors noted a few regions with negative correlation and mentioned those regions were removed in the analyses in the methods. Do the authors have explanations for such negative correlations?

Response: The proportion of identified regions with negative correlation (152 in 4160) was consistent with the FDR cutoff 0.05. Therefore, we considered the negatively correlated regions as possibly false positive findings and removed those regions from subsequent analysis.

6) The descriptions about inputs required to estimate $w_{\hat{}}$ is not consistent, i.e., whether summary statistics from validation cohorts are needed or not. (“Then, we provide detailed justifications on how to estimate....”, “Taken together, ... we only need the LD reference and summary statistics from a validation sample.”)

Response: We do not need the summary statistics from a validation sample. Instead, we introduced a novel repeated learning approach that allows estimating \hat{w} using only input GWAS summary statistics. We have clarified this as below (page 18 in the revised manuscript).

“Taken together, this shows that LD reference and summary statistics from a validation sample can be used to estimate \hat{w} . However, summary statistics from a validation cohort are still difficult to obtain in practice, and it is tempting to replace it with the input GWAS used for PRS training. But this is not feasible since it is a textbook example of overfitting. This motivates us to use repeated learning (or a similar cross-validation approach; see **Supplementary Note**)^{18,19} to estimate \hat{w} .”

7) In Figure S9 and S10, the improvement of X-Wing over PRS-CSx/XPASS seemed to vary with the number of regions selected. But the authors mentioned that “We selected $s = 1000$ in our primary analysis and demonstrated that PRS performance is robust to the choice of s ”, please use the same y-axis scales to present and run statistical test to see whether they are truly insignificant different.

Response: Thank for the suggestion. We used the two-sided Wilcoxon signed-rank test to compare the predictive R2 of PRS using varying number of top regions ($s = 500, 1000, \text{ or } 1500$). All Wilcoxon signed-rank test failed to reject the null hypothesis, indicating prediction accuracies of PRS with different choices of s are not statistically different (**Supplementary Table 19 and 20**).

Supplementary Table 19. P-value for comparing the prediction accuracy of X-Wing PRS with varying numbers of top regions for 31 traits in East Asian population. The prediction accuracy is evaluated using partial R2. P-value is from two-sided Wilcoxon signed-rank test.

PRS	Comparison		P-value
European PRS	Top 500 positive	Top 1000 positive	0.38
	Top 500 positive	Top 1500 positive	0.95
	Top 1000 positive	Top 1500 positive	0.56
Linearly combined PRS	Top 500 positive	Top 1000 positive	0.84
	Top 500 positive	Top 1500 positive	0.38
	Top 1000 positive	Top 1500 positive	0.29

Supplementary Table 20. P-value for comparing the prediction accuracy of X-Wing PRS with varying numbers of top regions for 13 traits in admixed American population. The prediction accuracy is evaluated using partial R2. P-value is from two-sided Wilcoxon signed-rank test.

PRS	Comparison		P-value
European PRS	Top 500 positive	Top 1000 positive	0.59
	Top 500 positive	Top 1500 positive	0.34
	Top 1000 positive	Top 1500 positive	0.84
East Asian PRS	Top 500 positive	Top 1000 positive	0.74
	Top 500 positive	Top 1500 positive	0.41
	Top 1000 positive	Top 1500 positive	0.89
Linearly combined PRS	Top 500 positive	Top 1000 positive	0.79
	Top 500 positive	Top 1500 positive	0.64
	Top 1000 positive	Top 1500 positive	0.79

We also have modified the original Figure S9 and S10 to have the same y-axis to better present the results (**Supplementary Figure 12 and 13**).

Supplementary Figure 12. Comparison of the prediction accuracy between X-Wing and PRS-CSx PRS for 31 traits in East Asians with varying numbers of top regions. Panel a, b, and c show the percentage increase in R^2 of X-Wing European PRS over PRS-CSx using 500, 1000, and 1500 positive regions as annotation. Panel d, e, and f are the results for linearly combined PRS. The dashed line represents the average increase.

Supplementary Figure 13. Comparison of the prediction accuracy between X-Wing and PRS-CSx PRS for 13 traits in admixed Americans with varying numbers of top regions. Panel a, d, and g show the percentage increase in R^2 of European PRS from X-Wing over PRS-CSx using 500, 1000, and 1500 positive regions as annotation. Panel b, e, and h are the results for East Asian PRS. Panel c, f, and i represent the results for linearly combined PRS. The dashed line represents the average increase.

8) The computation cost and requirements (e.g., whether can be run paralleled) is missing for X-Wing as compared to other methods, which would be helpful for future users.

Response: Thanks for the suggestions. We have added descriptions of the computation cost and requirements for X-Wing (page 18 in the **Supplementary Note**).

“X-Wing by default runs analysis on each chromosome and estimates the posterior effects for each population in parallel. The computational demands of X-Wing depend on the number of SNPs in GWAS summary statistics. For our analysis using UKBB and BBJ summary statistics as discovery GWAS and UKBB East Asian samples as test data, it takes about 1.5 hours to finish estimating posterior effects in chromosome 1 and about 10 minutes in chromosome 22 with 2000 MCMC iterations (default), using single thread in Intel Xeon Gold E5-4620 processor (2.60 GHz).

We further compared the computation cost and requirements between X-Wing, PRS-CSx, and XPASS using the same machine. XPASS does not allow parallel computation but takes the shortest time to finish the computing for all 22 chromosomes. Both PRS-CSx and X-Wing allow parallel computing over chromosomes. For the longest chromosome (chromosome1), PRS-CSx takes about 1 hour to finish the computation and has similar memory usage as X-Wing (**Supplementary Table 25**). Since X-Wing requires estimating the posterior effects for each population separately, the number of parallel computing jobs of X-Wing is K times of PRS-CSx, where K is the number of GWAS summary statistics included.”

9) How global genetic correlation was calculated here?

Response: Global genetic correlation was calculated using the XPASS software. We have added the description in the caption of **Figure 3**.

10) Are the individuals in HapGen2 all unrelated?

Response: The software HapGen2 simulates genotypes from reference haplotypes in 1000 Genomes Project. We calculated the pairwise identity by descent of simulated genotypes across 50000 Europeans. The mean value is 0.021 which is close to 0, indicating that the simulated genotypes for different individuals are approximately independent.

References

1. Zhang, H. *et al.* Novel Methods for Multi-ancestry Polygenic Prediction and their Evaluations in 3.7 Million Individuals of Diverse Ancestry. *bioRxiv* (2022).
2. Li, X. *et al.* Dynamic incorporation of multiple in silico functional annotations empowers rare variant association analysis of large whole-genome sequencing studies at scale. *Nature genetics* **52**, 969-983 (2020).
3. Amariuta, T. *et al.* Improving the trans-ancestry portability of polygenic risk scores by prioritizing variants in predicted cell-type-specific regulatory elements. *Nature genetics* **52**, 1346-1354 (2020).
4. Zhang, Q., Privé, F., Vilhjálmsson, B. & Speed, D. Improved genetic prediction of complex traits from individual-level data or summary statistics. *Nature communications* **12**, 1-9 (2021).
5. Weissbrod, O. *et al.* Leveraging fine-mapping and multipopulation training data to improve cross-population polygenic risk scores. *Nature Genetics* (2022).
6. Wainschtein, P. *et al.* Assessing the contribution of rare variants to complex trait heritability from whole-genome sequence data. *Nature Genetics* **54**, 263-273 (2022).
7. Li, X. *et al.* Dynamic incorporation of multiple in silico functional annotations empowers rare variant association analysis of large whole-genome sequencing studies at scale. *Nature Genetics* **52**, 969-983 (2020).
8. Zhou, H. *et al.* FAVOR: Functional Annotation of Variants Online Resource and Annotator for Variation across the Human Genome. *bioRxiv* (2022).
9. Zhao, Z. *et al.* PUMAS: fine-tuning polygenic risk scores with GWAS summary statistics. *Genome biology* **22**, 1-19 (2021).
10. Shi, H., Mancuso, N., Spendlove, S. & Pasaniuc, B. Local genetic correlation gives insights into the shared genetic architecture of complex traits. *The American Journal of Human Genetics* **101**, 737-751 (2017).
11. Zhang, Y. *et al.* SUPERGNOVA: local genetic correlation analysis reveals heterogeneous etiologic sharing of complex traits. *Genome biology* **22**, 1-30 (2021).
12. Henderson, C.R. Best linear unbiased estimation and prediction under a selection model. *Biometrics*, 423-447 (1975).
13. De Los Campos, G., Gianola, D. & Allison, D.B. Predicting genetic predisposition in humans: the promise of whole-genome markers. *Nature Reviews Genetics* **11**, 880-886 (2010).
14. Hu, Y. *et al.* Leveraging functional annotations in genetic risk prediction for human complex diseases. *PLoS computational biology* **13**, e1005589 (2017).
15. Márquez-Luna, C. *et al.* Incorporating functional priors improves polygenic prediction accuracy in UK Biobank and 23andMe data sets. *Nature Communications* **12**, 1-11 (2021).
16. Scott, R.A. *et al.* An Expanded Genome-Wide Association Study of Type 2 Diabetes in Europeans. *Diabetes* **66**, 2888-2902 (2017).
17. Suzuki, K. *et al.* Identification of 28 new susceptibility loci for type 2 diabetes in the Japanese population. *Nature Genetics* **51**, 379-386 (2019).
18. Allen, D.M. The relationship between variable selection and data augmentation and a method for prediction. *technometrics* **16**, 125-127 (1974).

19. Bates, S., Hastie, T. & Tibshirani, R. Cross-validation: what does it estimate and how well does it do it? *arXiv preprint arXiv:2104.00673* (2021).

REVIEWER COMMENTS

Reviewer #1 (Remarks to the Author):

The authors fully addressed the reviewer's comments. This reviewer appreciates the efforts to expand the simulation studies.

Reviewer #2 (Remarks to the Author):

In this updated paper, Miao et al has provided an updated to their previous manuscript which answered some of my questions. Thank you for the effort. However, maybe due to a slight difference in background, I still struggle to understand a few concepts:

1. In the rebuttal letter, the authors provide a detail step by step breakdown of their method, which is helpful for my understanding, however, I failed to understand how the authors calculate the polygenic risk score on the summary statistics data, without the sample genotype. Shouldn't a PRS refer to a per sample score?
2. This was brought up by reviewer 1 and 3. There are increasing number of GWAS performed using the linear mix model where the related samples might be included in the GWAS. My worry with this is that when performing the cross validation procedure, there might be data leakage in the cross validation, i.e. some samples in the training data are related to the validation sample, which might lead to bias in the cross validation procedure.
3. Sorry if I have missed this in previous review: For figure 3, there are some traits with genetic correlation > 1 (even when considering the confidence interval). Is there any explanation to that?

Reviewer #3 (Remarks to the Author):

I appreciate the efforts made by the authors to address my comments. However, I still have a few concerns:

1. I personally think the novelty of no requirement for validation cohort to calculate PRS weights is overstated as similar work has been mentioned in authors' previous work (Zhao et al. Genome Biology, 2021). Also, the point of not comparing linearly combined PRS using PRS-CSx with X-Wing is not totally convincing in Figure 5 and elsewhere. First, users can always split the target population itself to obtain both validation and test cohort. Second, there is an additional option `--meta` in PRS-CSx thus no additional validation cohort is required.

2. Related to my previous comments: Overall, I think the way how the results were presented might be misleading. Throughout the main manuscript, only the percentage of relative improvement was presented, without standard errors or p-values etc. In all comparisons, the absolute improvement was mostly slight. For example, BAS claimed to show the highest relative improvement while the partial R2 is only 0.007 VS 0.005 for X-Wing and PRS-CSx, respectively. Also, please add specific values and revise it as relative improvement when mentioning "basophil count showed the highest R2 improvement". I understand that the authors have added Wilcoxon test to compare the overall performance across traits between different methods, however, I think, similar statistics should be presented for each trait comparison or else absolute R2 differences (with error bars) instead should be reported.

3. I am not sure whether there is a widely accepted definition of PRS portability or generalizability. However, it is more reasonably to interpret it as the relative accuracy between EUR and non-EUR when using EUR-based PRS. So, I think the results showed the difference of cross-population prediction accuracy between methods, but no PRS portability was actually reported in this manuscript.

4. Also related to my previous comment: There is no one-size-fits-all method, how the performance of X-Wing will be affected by trait-specific genetic architecture (e.g., heritability, polygenicity) or other factors? It would be very helpful if the authors can summarize best practice or comments for using X-Wing. For example, would X-Wing show its superiority for traits such as BAS when there is low genome-wide r_g between populations; should total GWAS sample size or effective sample size used for binary traits given the marginal effects assumption etc.

Reviewer #2 (Remarks to the Author):

In this updated paper, Miao et al has provided an updated to their previous manuscript which answered some of my questions. Thank you for the effort. However, maybe due to a slight difference in background, I still struggle to understand a few concepts:

1. In the rebuttal letter, the authors provide a detail step by step breakdown of their method, which is helpful for my understanding, however, I failed to understand how the authors calculate the polygenic risk score on the summary statistics data, without the sample genotype. Shouldn't a PRS refer to a per sample score?

Response: Thank you for the comment. In our manuscript, we

- used the sample genotype to calculate the PRS when benchmarking and comparing the R^2 for different PRS methods
- did not use the sample genotype to calculate the PRS when estimating the local genetic correlation, SNP posterior effects, and linear combination weights for multiple PRS. These tasks only required GWAS summary statistics.

More specifically, we think the reviewer's question is referring to step 4 in our toy example to explain summary statistics-based cross-validation, "4. Generate population-specific PRS in the 25% validation sets and estimate the linear combination weights.", in our previous response letter.

Indeed, we do not need to calculate the PRS using sample genotype to estimate the linear combination weights. The idea is that the GWAS summary statistics and LD matrix are sufficient statistics for the least squares estimates of the linear combination weights.

Next, we provide a detailed mathematical derivation using our toy example. The model to estimate the linear combination weights is

$$Y^{(v)} \sim w_1 PRS_1^{(v)} + w_2 PRS_2^{(v)},$$

where $Y^{(v)}$ is the $N^{(v)}$ -dimensional phenotype vector in the 25% validation set, and $PRS_1^{(v)} = X^{(v)}b_1$ and $PRS_2^{(v)} = X^{(v)}b_2$ are the $N^{(v)}$ -dimensional population-specific PRS vector, w_1 and w_2 are the linear combination weights. We can rewrite our model into matrix form by

$$Y^{(v)} \sim PRS^{(v)}w,$$

where $PRS^{(v)} = [PRS_1^{(v)}, PRS_2^{(v)}] = X^{(v)}b$, b is the SNP effects used to calculate PRS, $w = (w_1, w_2)^T$. Then, we have the least squares estimates of the linear combination weights $w = (w_1, w_2)^T$ is

$$\begin{aligned}\hat{w} &= [PRS^{(v)T} PRS^{(v)}]^{-1} PRS^{(v)T} Y^{(v)} \\ &= [b^T X^{(v)T} X^{(v)} b]^{-1} b^T X^{(v)T} Y^{(v)} \\ &= \left[N^{(v)} b^T \frac{X^{(v)T} X^{(v)}}{N^{(v)}} b \right]^{-1} b^T X^{(v)T} Y^{(v)}\end{aligned}$$

$$\approx \left[N^{(v)} \mathbf{b}^T \frac{\mathbf{X}^{(ref)T} \mathbf{X}^{(ref)}}{N^{(ref)}} \mathbf{b} \right]^{-1} \mathbf{b}^T \mathbf{X}^{(v)T} \mathbf{Y}^{(v)}$$

The approximation comes from using LD matrix from reference panel $\frac{\mathbf{X}^{(ref)T} \mathbf{X}^{(ref)}}{N^{(ref)}}$ to replace the in-sample LD matrix $\frac{\mathbf{X}^{(v)T} \mathbf{X}^{(v)}}{N^{(v)}}$.

Therefore, we do not need to calculate the PRS using sample genotype data to estimate the linear combination weights. Instead, we only require LD matrix $\frac{\mathbf{X}^{(ref)T} \mathbf{X}^{(ref)}}{N^{(ref)}}$, the GWAS summary statistics $\mathbf{X}^{(v)T} \mathbf{Y}^{(v)}$, and SNP effects \mathbf{b} (e.g., the SNP posterior effects from any Bayesian PRS methods) to estimate the linear combination weights.

We have also added the detailed derivation of the linear combination weights in the Methods section of our manuscript (Page 19).

2. This was brought up by reviewer 1 and 3. There are increasing number of GWAS performed using the linear mix model where the related samples might be included in the GWAS. My worry with this is that when performing the cross validation procedure, there might be data leakage in the cross validation, i.e. some samples in the training data are related to the validation sample, which might lead to bias in the cross validation procedure.

Response: We thank the reviewer for this comment. The empirical impact of potential data leakage when applying summary statistics-based cross-validation to LMM summary statistics has not been sufficiently studied in the literature. However, as long as the testing dataset we use to evaluate R^2 is independent from the training data, then the model performance can be fairly assessed and compared. In our paper, the X-Wing linear combination approach based on GWAS summary statistics showed nearly identical results compared to the linearly combined PRS using the gold-standard regression approach trained on independent, individual-level data (**Figure 5b**, attached for convenience). This indicates that the empirical prediction accuracy was not affected by the fact that input GWAS associations were obtained from LMM on related samples, demonstrating the robustness of our approach to this issue.

Figure 5. Performance of X-Wing in combining population-specific PRS using GWAS summary statistics for 31 traits in East Asian samples. (b) Comparison of R^2 for linearly combined PRS with mixing weights obtained using GWAS summary statistics and individual-level data. The X-axis represents the R^2 using weights estimated from individual-level data, while the Y-axis shows the R^2 using summary statistics-based weights. The dashed line represents the diagonal line of $y=x$.

Furthermore, we have added an analysis using UK Biobank data to compare our summary statistics-based cross-validation on LMM summary statistics with the gold-standard individual data-based cross-validation done in independent samples. We used the LMM summary statistics of a randomly selected sample of 250,000 Europeans (with related samples) as input to our method. A random sample of 250,000 independent European individuals was used for the individual data-based cross-validation. We trained X-Wing using the European GWAS in these two designs and East Asian GWAS summary statistics from Biobank Japan for height. Our goal is to obtain a linear combination of European and East Asian PRS to predict phenotype in European samples. We conducted the analysis using 75% sample as training data and the remaining 25% samples as validation data. We found that the linear combination weights are highly consistent between these two designs across 4 replicates (**Response letter Fig 1** below). This result once again indicates that our summary-statistics-based approach is robust to the potential data leakage in LMM.

Response letter figure 1. Comparison of linear combination weights between summary statistics-based and individual data-based cross-validation. The European PRS relative linear combination weights is defined as (weights for European PRS)/(weights for European PRS + weights for East Asian PRS).

3. Sorry if I have missed this in previous review: For figure 3, there are some traits with genetic correlation > 1 (even when considering the confidence interval). Is there any explanation to that?

Response: Thank you for the comment. The error bars in Figure 3b represent standard errors of genetic correlation as stated in the figure caption. For your convenience, we also re-plotted the figure where error bars represent $1.96 \times$ standard errors, as shown below. Genetic correlation estimates in regions identified by X-Wing

are larger than 1 for three traits (i.e. LYM, AST, and SBP). However, the genetic correlation estimate is not significantly larger than 1 across all traits after considering the 95% confidence interval.

Response letter figure 2. Cross-population genetic correlation for 31 complex traits. The three bars denote the global genetic correlation estimated from genome-wide data (light green), genetic correlation in regions identified by X-Wing (brown), and genetic correlation outside regions identified by X-Wing (dark green). Results for a simulated uncorrelated trait are labeled as 'Control'. The traits are ordered according to the global genetic correlation estimates. Error bars indicate $\pm 1.96 * \text{standard errors}$.

Reviewer #3 (Remarks to the Author):

I appreciate the efforts made by the authors to address my comments. However, I still have a few concerns:

1. I personally think the novelty of no requirement for validation cohort to calculate PRS weights is overstated as similar work has been mentioned in authors' previous work (Zhao et al. Genome Biology, 2021). Also, the point of not comparing linearly combined PRS using PRS-CSx with X-Wing is not totally convincing in Figure 5 and elsewhere. First, users can always split the target population itself to obtain both validation and test cohort. Second, there is an additional option –meta in PRS-CSx thus no additional validation cohort is required.

Response: We thank the reviewer for the comment. The reviewer raised three concerns that we will address one by one:

1) Novelty of our summary statistic-based model integration and comparison with another paper from our group: Zhao et al. Genome Biology, 2021

--Zhao et al.¹ previously showed that selection of tuning parameters can be performed using summary statistics. That paper did not show any results on combining multiple PRS with GWAS summary statistics. To the best of our knowledge, this paper is the first to demonstrate that optimal linear combination of PRS can also be done using summary statistics alone. Given that combination of multiple ancestry-specific PRS provides a substantial gain in prediction accuracy in cross-ancestry PRS applications as demonstrated by many recent PRS papers using individual-level data²⁻⁴, this methodological advance represents a big step forward in the field.

2) The users can always split the target population itself to obtain both validation and test cohorts and use the validation cohort to do the model integration.

--Respectfully, we argue that in many real PRS applications it is impractical to use a part of the target samples for combining PRS models. For example, one important application of PRS is to identify high-risk individuals earlier in life and provide a basis for early intervention and treatment⁵. In this type of application, genetic data are available but not the phenotypes, which means the target population cannot be used for model integration. In many other studies, researchers are interested in producing PRS to quantify genetic predisposition for certain traits, then using these scores to test their hypothesis. One recent example is a study which assesses whether PRS modifies the effect of education on health outcomes⁶. In this type of application, it is crucial to use the whole target cohort in the downstream analysis in order to maximize statistical power. Reserving a portion of target sample just for model integration is far from ideal. The example given by the reviewer where the target population can be split in half mostly only applies to methodological development work where the researchers only want to benchmark the performance of different methods. We argue that being able to

implement PRS combination on summary statistics alone is a major advance that will benefit many real-world PRS applications.

3) PRS-CSx has the “-meta” option that requires no validation cohort.

--In theory, our way to linearly combine multiple PRS is statistically optimal as it minimizes the squared loss between the phenotype and the linearly combined PRS using least squares. In contrast, the "meta" option in PRS-CSx, which performs inverse-variance-weighted meta-analysis on SNP posterior effects, is not expected to be optimal for prediction tasks. In fact, the author of PRS-CSx stated that “We note that, although simpler to implement, the ‘meta’ option is expected to be less accurate compared with the linear combination approach that optimizes PRS estimation separately in each target population.”² Our summary statistic-based linear combination has shown to have comparable prediction accuracy to the individual-level data-based approach. Therefore, we anticipate that it will outperform the "meta" option in PRS-CSx.

To assess this, we have also added a comparison of our approach to the "meta" option in PRS-CSx. The results are shown in **Supplementary Figure 22** (also attached here for convenience). Our approach showed an average R^2 relative increase of 11.6% ($P_{\text{wilcoxon}} = 3.6e-3$), 11.9% ($P_{\text{wilcoxon}} = 0.017$), and 17.5% ($P_{\text{wilcoxon}} = 2.4e-4$) for traits in East Asians, Africans, and admixed Americans, respectively. This further demonstrates that our approach outperforms the “-meta” option in PRS-CSx.

Supplementary Figure 22. Comparison of the prediction accuracy between X-Wing and PRS-CSx “-meta” PRS. a) 31 traits in East Asians b) 13 traits in admixed Americans c) 13 traits in Africans. X-Wing PRS is based on summary statistics-based linear combined PRS. The dashed line represents the average increase.

We have also added texts to page 11 in the revised manuscript:

“We further compared X-Wing performance with the “-meta” option in PRS-CSx that requires no additional validation cohort. X-Wing showed an average R² relative increase of 11.6% ($P_{\text{wilcoxon}} = 3.6e-3$), 11.9% ($P_{\text{wilcoxon}} = 0.017$), and 17.5% ($P_{\text{wilcoxon}} = 2.4e-4$) for traits in East Asians, Africans, and admixed Americans, respectively.”

2. Related to my previous comments: Overall, I think the way how the results were presented might be misleading. Throughout the main manuscript, only the percentage of relative improvement was presented, without standard errors or p-values etc. In all comparisons, the absolute improvement was mostly slight. For example, BAS claimed to show the highest relative improvement while the partial R² is only 0.007 VS 0.005 for X-Wing and PRS-CSx, respectively. Also, please add specific values and revise it as relative improvement when mentioning “basophil count showed the highest R² improvement”. I understand that the authors have added Wilcoxon test to compare the overall performance across traits between different methods, however, I think, similar statistics should be presented for each trait comparison or else absolute R² differences (with error bars) instead should be reported.

Response: Thanks for the suggestion. We have added text to our manuscript to emphasize that while our method has shown an overall improved prediction accuracy across traits, the relative improvement in R^2 for a single trait may be statistically imprecise and should be interpreted with caution (see page 14 of the revised manuscript, also attached here for convenience). We have also added the standard errors of the percentage of relative improvement in **Supplementary Table 12**. Furthermore, we have removed the texts that highlight the improvement of R^2 for BAS following the reviewer's suggestion (Page 9 in the manuscript). Finally, we note that while we agree that presenting the relative R^2 does not show the full picture of PRS performance, it is a fairly commonly used approach to compare different PRS methods. In fact, this is how the PRS-CSx paper demonstrated its superior performance over other methods². We have also shown statistical significance results using Wilcoxon test (and now standard error for R^2 gain) which provides further evidence for the main conclusions in our paper.

Page 14 in the revised manuscript:

“Third, although we have demonstrated an overall improved prediction accuracy over alternative methods across many traits, the relative improvement in R^2 reported for a single trait may be statistically imprecise (**Supplementary Table 12**) and should be interpreted with caution.”

3. I am not sure whether there is a widely accepted definition of PRS portability or generalizability. However, it is more reasonable to interpret it as the relative accuracy between EUR and non-EUR when using EUR-based PRS. So, I think the results showed the difference of cross-population prediction accuracy between methods, but no PRS portability was actually reported in this manuscript.

Response: Thank you for the comment. Following your comment, we have revised the text to change the “portability” to “prediction accuracy” when comparing different methods.

4. Also related to my previous comment: There is no one-size-fits-all method, how the performance of X-Wing will be affected by trait-specific genetic architecture (e.g., heritability, polygenicity) or other factors? It would be very helpful if the authors can summarize best practice or comments for using X-Wing. For example, would X-Wing show its superiority for traits such as BAS when there is low genome-wide r_g between populations; should total GWAS sample size or effective sample size used for binary traits given the marginal effects assumption etc.

Response: We appreciate the comment. Overall, we considered the superior performance of X-Wing can be attributed to the incorporation of cross-population local genetic correlation and summary statistics-based PRS linear combination. Since our model is a mathematical generalization of PRS-CSx, we anticipate that the inclusion of cross-population local genetic correlation will lead to improved prediction accuracy. However, the estimation of local genetic correlation may be noisy with

small GWAS sample sizes which will impact the prediction accuracy. For summary statistics-based PRS linear combination, we recommend it in general due to its superior and robust performance. In most cases, the summary statistics from multiple populations are available and thus allow us to implement the PRS linear combination. If there are concerns about the quality of cross-population local genetic correlation, integrating summary statistics-based PRS linear combination into existing methods such as PRS-CSx² and PolyPred³ is still a valid strategy. For binary traits, we recommend using the effective sample size as described in Privé et al.⁷

We have added the best practice for using X-Wing in the discussion (page 14) of the revised manuscripts:

“Finally, the overall superior performance of X-Wing can be attributed to the incorporation of cross-population local genetic correlation and summary statistics-based PRS combination. Although we anticipate improved prediction accuracy after incorporating the local genetic correlation annotation, imprecise estimation of local genetic correlation may affect PRS performance when input GWAS have limited sample size. However, the summary statistics-based PRS combination strategy is highly robust in our analyses. In cases where there are concerns about the quality of local genetic correlation estimation, integrating summary statistics-based PRS combination into existing methods^{2,3} should still be a strategy for consideration. “

References

1. Zhao, Z. *et al.* PUMAS: fine-tuning polygenic risk scores with GWAS summary statistics. *Genome biology* **22**, 1-19 (2021).
2. Ruan, Y. *et al.* Improving polygenic prediction in ancestrally diverse populations. *Nature Genetics* (2022).
3. Weissbrod, O. *et al.* Leveraging fine-mapping and multipopulation training data to improve cross-population polygenic risk scores. *Nature Genetics* (2022).
4. Albiñana, C. *et al.* Leveraging both individual-level genetic data and GWAS summary statistics increases polygenic prediction. *The American Journal of Human Genetics* **108**, 1001-1011 (2021).
5. Kullo, I.J. *et al.* Polygenic scores in biomedical research. *Nature Reviews Genetics*, 1-9 (2022).
6. Barcellos, S.H., Carvalho, L.S. & Turley, P. Education can reduce health differences related to genetic risk of obesity. *Proceedings of the National Academy of Sciences* **115**, E9765-E9772 (2018).
7. Privé, F., Arbel, J., Aschard, H. & Vilhjálmsson, B.J. Identifying and correcting for misspecifications in GWAS summary statistics and polygenic scores. *Human Genetics and Genomics Advances* **3**(2022).

REVIEWERS' COMMENTS

Reviewer #2 (Remarks to the Author):

Thank you, the authors have addressed all my questions. Only comment is that I think the figure with the 95% confidence interval might be more appropriate when compared to the standard error for figure 3, as it is still rather confusing as to why one can obtain a genetic correlation larger than 1 when the error bar does not cover 1

Reviewer #3 (Remarks to the Author):

The authors have satisfactorily addressed most of my comments. I appreciate the additional efforts by adding the results of PRS-CSx using --meta option, although there might be some caveats here regarding the inverse-variance weighted meta-analysis versus linear combination. I think for traits with large effect ancestry-specific variants or low cross-ancestry genetic correlation, the weighted strategy would perform better than meta. In other cases, if the causal variants are largely shared across populations, especially for causal variants, I don't see the harm using meta strategy. Also, the sample size ratio of GWAS from multiple populations could affect the performance of meta and weighted strategy. Nevertheless, for what it has done, the analysis is comprehensive, and the research question explored here is important and interesting. Another minor point is that median accuracy is more suitable to report if you are performing Wilcoxon test.

Reviewer #2 (Remarks to the Author):

Thank you, the authors have addressed all my questions. Only comment is that I think the figure with the 95% confidence interval might be more appropriate when compared to the standard error for figure 3, as it is still rather confusing as to why one can obtain a genetic correlation larger than 1 when the error bar does not cover 1

Response: Thank you for the comment. We have modified Figure 3 such that the error bar represents the 95% confidence interval.

Reviewer #3 (Remarks to the Author):

The authors have satisfactorily addressed most of my comments. I appreciate the additional efforts by adding the results of PRS-CSx using --meta option, although there might be some caveats here regarding the inverse-variance weighted meta-analysis versus linear combination. I think for traits with large effect ancestry-specific variants or low cross-ancestry genetic correlation, the weighted strategy would perform better than meta. In other cases, if the causal variants are largely shared across populations, especially for causal variants, I don't see the harm using meta strategy. Also, the sample size ratio of GWAS from multiple populations could affect the performance of meta and weighted strategy. Nevertheless, for what it has done, the analysis is comprehensive, and the research question explored here is important and interesting. Another minor point is that median accuracy is more suitable to report if you are performing Wilcoxon test.

Response: Thank you for the comment. Following your comment, we have revised the texts to report the median relative improvement in predictive accuracy.